# Multivalent interactions essential for lentiviral integrase function

Allison Ballandras-Colas [1,14,18], Vidya Chivukula [1,15,18], Dominika T. Gruszka [2,16], Zelin Shan [3], Parmit K. Singh [4,5], Valerie E. Pye [1], Rebecca K. McLean [6,17], Gregory J. Bedwell[4,5], Wen Li[4,5], Andrea Nans [7], Nicola J. Cook [1], Hind J. Fadel[8], Eric M. Poeschla [9], David J. Griffiths [6], Javier Vargas[10], Ian A. Taylor [11], Dmitry Lyumkis [3,12✉], Hasan Yardimci [2✉], Alan N. Engelman [4,5✉] & Peter Cherepanov [1,13✉]

A multimer of retroviral integrase (IN) synapses viral DNA ends within a stable intasome nucleoprotein complex for integration into a host cell genome. Reconstitution of the intasome from the maedi-visna virus (MVV), an ovine lentivirus, revealed a large assembly containing sixteen IN subunits[1]. Herein, we report cryo-EM structures of the lentiviral intasome prior to engagement of target DNA and following strand transfer, refined at 3.4 and 3.5 Å resolution, respectively. The structures elucidate details of the protein-protein and protein-DNA interfaces involved in lentiviral intasome formation. We show that the homomeric interfaces involved in IN hexadecamer formation and the α-helical configuration of the linker connecting the C-terminal and catalytic core domains are critical for MVV IN strand transfer activity in vitro and for virus infectivity. Single-molecule microscopy in conjunction with photobleaching reveals that the MVV intasome can bind a variable number, up to sixteen molecules, of the lentivirus-specific host factor LEDGF/p75. Concordantly, ablation of endogenous LEDGF/p75 results in gross redistribution of MVV integration sites in human and ovine cells. Our data confirm the importance of the expanded architecture observed in cryo-EM studies of lentiviral intasomes and suggest that this organization underlies multivalent interactions with chromatin for integration targeting to active genes.

[1] Chromatin Structure and Mobile DNA Laboratory, The Francis Crick Institute, London, UK. [2] Single Molecule Imaging of Genome Duplication and Maintenance Laboratory, The Francis Crick Institute, London, UK. [3] Laboratory of Genetics, The Salk Institute for Biological Studies, La Jolla, CA, USA. [4] Department of Cancer Immunology & Virology, Dana-Farber Cancer Institute, Boston, MA, USA. [5] Department of Medicine, Harvard Medical School, Boston, MA, USA. [6] Moredun Research Institute, Pentlands Science Park, Bush Loan, Penicuik, UK. [7] Structural Biology Science Technology Platform, The Francis Crick Institute, London, UK. [8] Division of Infectious Diseases, Mayo Clinic, Rochester, MN, USA. [9] Division of Infectious Diseases, University of Colorado Anschutz Medical Campus, Aurora, CO, USA. [10] Departmento de Óptica, Universidad Complutense de Madrid, Madrid, Spain. [11] Macromolecular Structure Laboratory, The Francis Crick Institute, London, UK. [12] Department of Integrative Structural and Computational Biology, The Scripps Research Institute, La Jolla, CA, USA. [13] Department of Infectious Disease, St-Mary's Campus, Imperial College London, London, UK. [14] Present address: Institut de Biologie Structurale (IBS) CNRS, CEA, University Grenoble, Grenoble, France. [15] Present address: Department of Microbiology, NYU Grossman School of Medicine, New York, NY 10016, USA. [16] Present address: Biological Physics Research Group, Clarendon Laboratory, Department of Physics and Kavli Institute for Nanoscience Discovery, University of Oxford, Oxford, UK. [17] Present address: The Pirbright Institute, Ash Road, Pirbright, Woking GU24 0NF, UK. [18] These authors contributed equally: Allison Ballandras-Colas, Vidya Chivukula. ✉email: dlyumkis@salk.edu; Hasan.Yardimci@crick.ac.uk; Alan_Engelman@dfci.harvard.edu; Peter.Cherepanov@crick.ac.uk

The *Retroviridae* family contains six orthoretroviral genera: alpha-, beta-, gamma-, delta-, epsilon-retroviruses, and lentiviruses, as well as a separate subfamily of spuma-viruses. Lentiviruses include human immunodeficiency virus type 1 (HIV-1), which is responsible for the global AIDS pandemic. Complementary research into non-human retroviral species greatly accelerated anti-HIV/AIDS drug development from the beginning of the pandemic[1]. Retroviral infection proceeds through reverse transcription of the viral RNA genome into a linear double-stranded viral DNA (vDNA) copy, which is then integrated into a host cell chromosome. Retroviral integrase (IN), the enzyme responsible for this process, catalyzes two consecutive reactions: (i) 3′-processing, during which IN hydrolyses 2–3 nucleotides at the vDNA ends to liberate 3′-hydroxyls attached to invariant CA dinucleotides, and (ii) strand transfer, wherein IN utilizes the 3′-hydroxyls to cleave the chromosomal DNA, simultaneously joining the 3′ vDNA ends to target DNA (tDNA) strands (reviewed in Ref. [2]). Both reactions proceed via $S_N2$ transesterification at phosphorus atoms and require a pair of divalent metal cations ($Mg^{2+}$ or $Mn^{2+}$) as cofactors[3,4].

To catalyze integration, a multimer of IN assembles into a nucleoprotein complex with synapsed vDNA ends, termed the intasome[5,6]. Results of detailed in vitro biochemical and structural studies suggest that the intasome assembles on non-processed vDNA ends as the initial synaptic complex (ISC) that sequentially transitions into the cleaved synaptic complex (CSC, upon 3′-processing of vDNA ends), target capture complex (TCC, upon tDNA binding), and, finally, the post-catalytic strand transfer complex (STC)[4]. Disassembly of the STC and subsequent joining of the 5′ vDNA ends to tDNA are thought to depend on cellular machineries.

Retroviral INs contain three canonical domains connected by highly divergent linkers. The amino-terminal domain (NTD) consists of a compact three-helical bundle stabilized by coordination of a $Zn^{2+}$ ion; the catalytic core domain (CCD) features the RNaseH fold and harbors the invariant D,D-35-E catalytic triad; and the carboxy-terminal domain (CTD) adopts an SH3-like beta-barrel structure (reviewed in Jaskolski et al.[7]). During the last decade, intasomes from five retroviral genera as well as from a yeast retroelement have been structurally characterized in their ISC, CSC, TCC, and/or STC forms[4,6,8–16]. They all share a common functional unit, termed the conserved intasomal core (CIC) assembled around a pair of vDNA ends. The CIC contains two IN CCD dimers, each providing one active site, joined by the exchange of a pair of NTDs. The two halves of the CIC are separated by a pair of CTDs, which act as rigid spacers separating the inner (catalytic) CCDs. The outer CCDs of the CIC do not play a catalytic function. Depending on the viral species, the CIC is decorated by a variable number of IN chains, completing the respective intasome assemblies. Whereas the intasome from the prototype foamy virus (PFV, a spumavirus) comprises a minimal tetrameric IN complex, representing little more in excess of the CIC[6,9], its lentiviral counterparts can contain as many as sixteen IN chains. Thus, in vitro assembly of intasomes from HIV-1 and the closely related red-capped mangabey simian immunodeficiency virus ($SIV_{rcm}$) yielded highly polydisperse populations containing 10-mer, 12-mer, and 16-mer species in various fractions[13,17,18]. By contrast, the intasome from maedi-visna virus (MVV, an ovine lentivirus) behaved as a near-homogenous population with a large majority of particles harboring four IN tetramers[8]. The smaller HIV-1 and $SIV_{rcm}$ IN-vDNA nucleoprotein complexes could be explained as fragments of the hexadecameric assembly observed in the MVV structure, due to a loss or partial disorder of individual IN subunits[17]. These results underscore the utility of MVV intasomes as a model for in vitro studies of lentiviral integration.

HIV-1 IN, as well as its counterparts from other lentiviral species, interact with the chromatin-associated host protein LEDGF/p75, which strongly enhances their in vitro strand transfer activity[8,19,20]. LEDGF/p75 and, to a lesser degree, its paralog HRP2, are largely responsible for the propensity of lentiviruses to integrate into active transcription units (TUs, reviewed in Bedwell and Engelman[21]). While causing variable and usually modest defects in integration efficiency, ablation of LEDGF/p75 in infected cells results in profound changes to the genomic distribution of HIV-1 integration sites[22–26]. LEDGF/p75 contains two compact domains connected by a long flexible linker. The N-terminal PWWP domain binds nucleosomes carrying trimethylated histone H3 Lys36 (H3K36me3)[27–29]. The IN-binding domain (IBD), which is located close to the C-terminus of the protein[30,31], binds the IN CCD dimerization interface, with additional salt bridge contacts between the IBD and the IN NTD stabilizing the interaction[32,33]. The modular organization is thought to allow LEDGF/p75 to act as a tether between IN and H3K36me3-containing nucleosomes, which are enriched within TUs and are associated with transcriptional elongation and pre-mRNA splicing activity[34–36]. Indeed, HIV-1 integration is strongly biased towards genes enriched for H3K36me3[37].

The biological significance of the expanded intasome architecture observed in lentiviruses remains unclear. If four IN molecules are sufficient to assemble a functional spumaviral[6] or deltaretroviral[14,15] intasome, why should a lentivirus require many more IN chains to support an equivalent mechanism? Here, we describe high-resolution structures of the MVV intasome in two functional states: (i) prior to engagement of tDNA and (ii) following strand transfer. The structures reveal fine details of the protein-protein and protein-DNA interactions within the lentiviral intasome, which could not be resolved in the previous cryo-EM reconstructions[8]. Using site-directed mutagenesis, we demonstrate that the hexadecameric assembly is essential for MVV IN strand transfer activity and virus infectivity. We also show that the MVV intasome can recruit as many as 16 LEDGF/p75 molecules, which may allow it to form multivalent interactions with chromatin, potentially allowing it to be more sensitive to the epigenetic status of target chromatin.

## Results

**High-resolution cryo-EM structures of the MVV intasome in two functional states.** We collected single-particle cryo-EM data on the MVV intasome assembled in the form of the CSC and purified under conditions used in our previous study[8]. The reconstruction was refined to an overall resolution of 3.4 Å (Supplementary Figs. 1a and 2a, Supplementary Table 1). Although MVV IN strand transfer activity and intasome assembly in vitro are strongly stimulated by LEDGF/p75, the host factor dissociates during intasome purification[8]. In agreement with the previously reported MVV CSC structure, which was refined to ~5 Å resolution, the present reconstruction lacked features consistent with LEDGF/p75 (Supplementary Fig. 2a). To visualize tDNA binding and to potentially enhance host factor retainment, we assembled the STC intasome using a branched DNA construct mimicking a product of strand transfer and purified it in a buffer with reduced salt concentration. We acquired cryo-EM images of the MVV STC intasome particles and refined the resulting reconstruction to an overall resolution of 3.5 Å (Supplementary Figs. 1b, 2a, and 3 and Supplementary Table 1). The well-resolved portions of the cryo-EM map encompassed the intasome with 30 base pairs (bp) of tDNA and two copies of the LEDGF/p75 IBD (Fig. 1a, d and Supplementary Fig. 2a). Because unstructured and flexible regions form the bulk of the LEDGF/p75 molecule[30], it was unsurprising that only the IBD, which directly engages IN, was observed in the map.

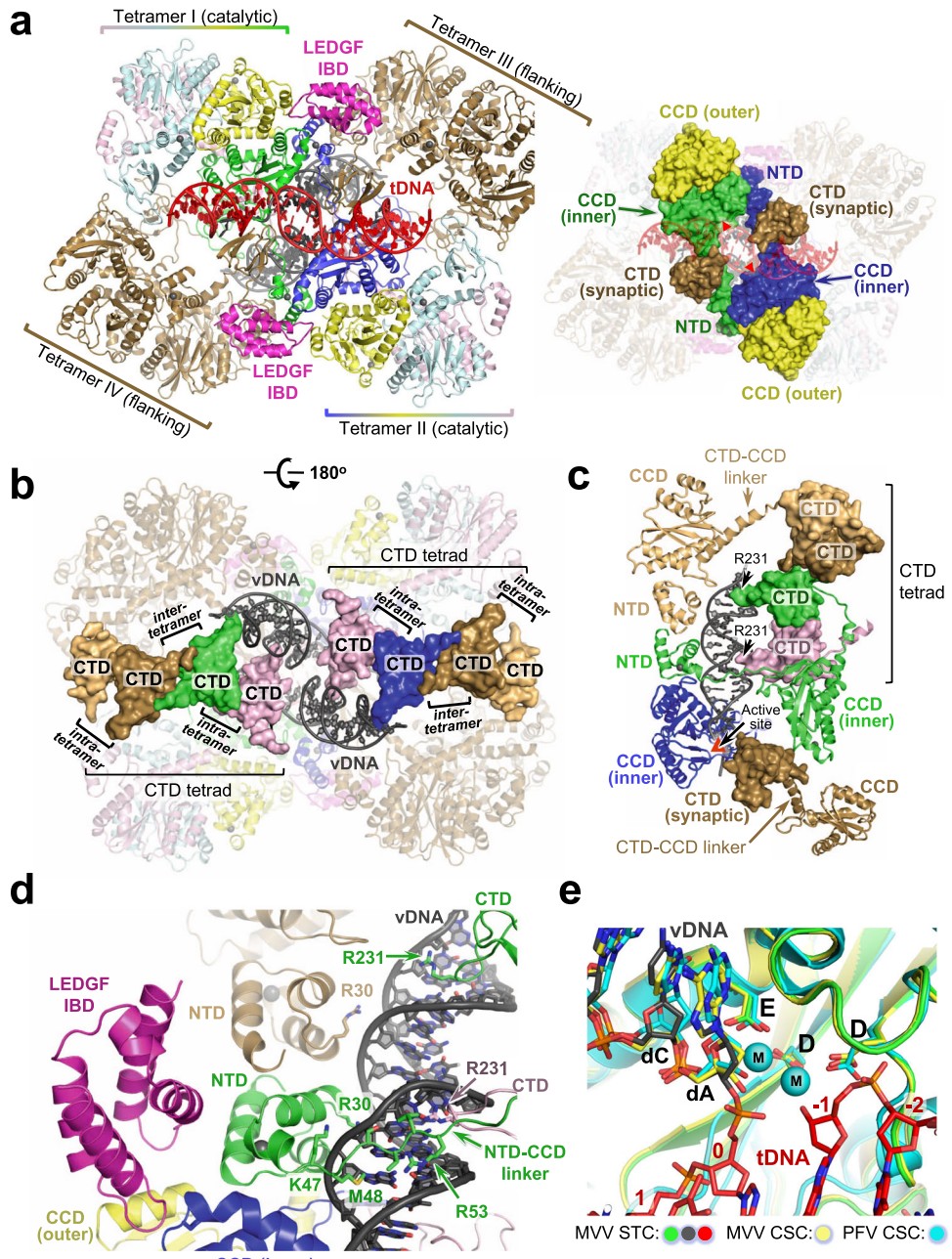

**Fig. 1 Overview of the MVV intasome architecture. a** Left: Refined model of the STC, shown as cartoons and color-coded to highlight LEDGF/p75 and IN subunits. Catalytic IN tetramers (tetramers I and II) are colored by subunit: the IN chains providing active sites are blue and green, and the remaining chains of the catalytic tetramers are shown in yellow, cyan, and light pink. The eight IN subunits comprising flanking tetramers III and IV are brown. LEDGF/p75 IBDs are magenta; vDNA and tDNA are gray and red, respectively. Right: The STC with IN domains contributing to the CIC shown in surface mode and indicated, and the remainder of the structure shown in semi-transparent cartoons. Red triangles depict active sites. **b** The STC with the CTDs comprising the tetrads in surface mode. **c** IN-vDNA interactions. One of the two vDNA ends is shown. Locations of IN domains (NTD, CCD, and CTD), CCD-CTD linkers, Arg231 residues involved in vDNA binding and the active site (red triangle) are indicated. **d** Closeup view of one of the LEDGF/p75 IBDs identified within the STC reconstruction. IN residues involved in the interactions with vDNA are shown as sticks and indicated. **e** Closeup view of the MVV STC active site region with IN, vDNA, and tDNA in green, gray, and red, respectively. Also shown are superposed structures of MVV CSC (yellow) and PFV CSC (cyan, PDB ID 3OY9)[7]. The three structures were superposed by the Cα atoms of the residues comprising the invariant D,D-35-E motif in each active site (indicated as D, D, and E, corresponding to Asp66, Asp118, and Glu154 in MVV, and Asp128, Asp185, and Glu221 in PFV IN). The protein and DNA are shown as cartoons and sticks, respectively; spheres (M) are catalytic $Mn^{2+}$ cations in the PFV crystal structure[7]. Residues of the invariant 3′ vDNA dCdA dinucleotide are indicated, and nucleotides of the tDNA in the MVV STC structure are numbered (0 corresponds to the nucleotide joined to 3′ end of vDNA).

The CSC and STC structures recapitulated the MVV intasome architecture harboring sixteen IN subunits in a tetramer-of-tetramers arrangement. Two IN chains provide active sites for catalytic function, while the remaining subunits are involved in protein-DNA and/or protein-protein interactions. Each of the four IN tetramers making up the intasome contributed to the formation of the CIC, which was resolved to 2.9–3.0 Å resolution within our cryo-EM reconstructions (Supplementary Figs. 2a and 3). The CIC

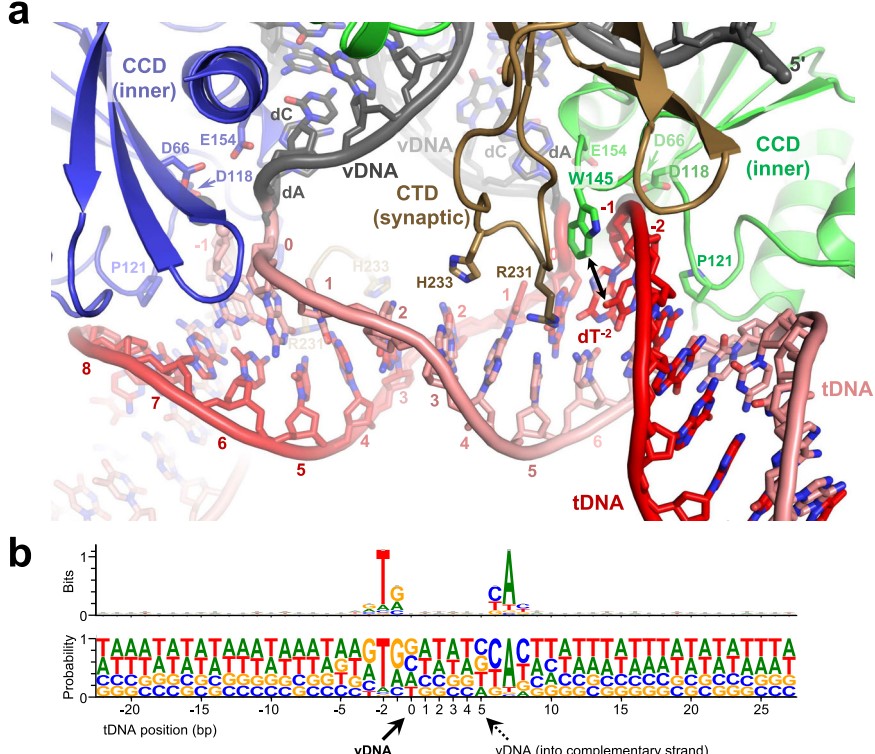

**Fig. 2 Engagement of tDNA by the MVV intasome. a** Closeup view of the vDNA-tDNA synapse within the MVV STC. DNA phosphate backbone is shown as cartoon with sugar and bases as sticks. Complementary tDNA strands are in red and pink. The rest of the chains are colored as in Fig. 1a. MVV IN residues involved in the interactions with tDNA are shown as sticks and indicated. **b** Nucleotide preferences at MVV integration sites. Sequence logos[111,113] represent information content (reported in bits, top) or raw nucleotide frequencies (bottom) at each position within an alignment of 327,911 in vitro MVV intasome integration sites in sheep genomic DNA[8]. Nucleotide positions of the tDNA strand represented by the logos are numbered, and the nucleotide that becomes joined to 3′ vDNA end (corresponding to position 0) is indicated with solid black arrowhead. The dotted arrowhead indicates the insertion position of the second vDNA end into the complementary tDNA strand.

is formed from a pair of CCD dimers provided by IN tetramers I and II, plus a pair of synaptic CTDs derived from IN tetramers III and IV (Fig. 1a). As in all other characterized intasomes, the inner (catalytic) IN chains of the CIC exchange their NTDs across the synaptic interface[2], with the entire extended NTD-CCD linker resolved in the cryo-EM maps (Supplementary Fig. 3b).

Comparison of the CSC and STC structures confirmed that the intasome does not undergo extensive conformational changes upon tDNA binding and strand transfer (Supplementary Fig. 2b). In agreement with other retroviral intasome structures[9,12,13,15], the tDNA bound the CIC within the cleft between a pair of catalytic IN CCDs, where it made additional contacts with the synaptic CTDs (Figs. 1a and 2a and Supplementary Fig. 3c). Considerable distortion of the duplex structure allowed placement of the tDNA scissile phosphodiester bonds, separated by 20 Å in B-form DNA, into the intasomal active sites spaced by 30 Å. Compared to non-lentiviral species, the MVV intasome induced less extensive tDNA bending into the minor groove, resulting in an obtuse angle between the tDNA arms and a lack of minor groove compression (Supplementary Fig. 4). MVV achieves the required widening of the tDNA major groove by more extensive stretching and underwinding of the duplex (Supplementary Fig. 4), resulting in a 4 Å stretch and 22° undertwist of the 6-bp tDNA segment between insertion sites of the vDNA ends. The distortion of the tDNA duplex is stabilized by interactions with IN subunits comprising the CIC. Two pairs of MVV IN Arg231 and His233 residues, located on the synaptic CTDs, along with Trp145 on the inner CCDs of the CIC, make direct contacts within the expanded tDNA major groove (Fig. 2a).

His233 is within hydrogen bonding distance of the tDNA backbone, while the side chains of Arg231 and Trp145 are in proximity with the C5 methyl group of the deoxythymidine at tDNA position −2. The hydrophobic interactions in the major groove may contribute to the strong preference for a thymine base at this position of MVV integration sites and the symmetric preference for adenine at position +7 (Fig. 2b)[8]. Notably, Arg231 is structurally equivalent to PFV IN Arg329, which likewise packs within the major groove of tDNA and aids its deformation[9].

During the strand transfer reaction, the tDNA scissile phosphodiester group is coordinated by the catalytic metal ion pair in the IN active site[4]. Crystallographic studies of PFV intasomes revealed that strand transfer leads to ejection of the trans-esterified phosphodiester from the active site[4,9], which is thought to drive the reaction towards integration by discouraging the reversal of transesterification. Although we acquired cryo-EM data in the absence of $Mg^{2+}$ or $Mn^{2+}$ ions, superposition of the structures with the active site region of the PFV intasome, which was crystallized in the presence of $Mn^{2+}$, allowed modeling the catalytic metal ions in the MVV IN active site (Fig. 1e). Within the MVV STC, the phosphodiester linking viral and target DNA molecules is 6 Å from the predicted position of either metal ion, indicating that reconfiguration of the active site following strand transfer is conserved in lentiviral intasomes.

**The CTD-CTD interfaces and the α-helical configuration of the CCD-CTD linker are critical for MVV IN strand transfer activity.** Retroviral intasomes display striking architectural diversity[2],

and outside of the CIC the MVV intasome is distinct from all characterized non-lentiviral intasomes. The cryo-EM structures revealed a critical role played by IN CTDs to stabilize the entire hexadecameric MVV intasome assembly. The CTDs from eight IN subunits form a pair of C2 symmetry-equivalent tetrads stabilizing the hexadecameric assembly via intra- and inter-tetramer CTD-CTD interactions (Fig. 1b) as well as via contacts with vDNA (Fig. 1c). CTD dimerization is conserved among lentiviral INs[8,13,17,38], although further multimerization into the tetrad has thus far been observed only for MVV IN in the context of the vDNA-bound intasome[8]. Another characteristic feature, unique to lentiviral INs, is the α-helical configuration of the linker connecting the CTD and CCD[8,39]. Importantly, the CTD-CTD interfaces and CCD-CTD linker do not directly contribute to the CIC. Although the CIC contains a pair of CTDs, these assume unique synaptic positions and do not participate in homomeric CTD-CTD interactions.

To test the functional significance of the lentivirus-specific MVV intasome features, we targeted select IN residues by site-directed mutagenesis. Amino acid substitutions F223A, Y225A, W245E/L, V252A/D, Y261A/E, V263E, and I272E were designed to disrupt MVV IN intra- and inter-tetramer CTD-CTD interfaces (Fig. 3a, left). To destabilize the CCD-CTD linker (Fig. 3a, right), we produced MVV IN variants harboring substitutions of Gln residues for helix-destabilizing Pro and Gly (QQ207GP and QQQ211PGG). To the same end, we constructed a chimera carrying a portion of the PFV IN CCD-CTD linker (HPSTPPASSRS, corresponding to PFV IN residues 304–314, known to adopt an extended conformation[6]) inserted between MVV IN Gln210 and Gln214, resulting in the variant referred to as QQQ211PFV$_{304-314}$. As controls, we produced MVV IN E154Q, R231E, and H12N variants. Glu154, which is the catalytic glutamate of the IN D,D-35-E motif (Fig. 1e), does not contribute to any of the IN-IN or IN-DNA interfaces of the intasome. Therefore, E154Q was expected to abrogate IN catalytic activity, while preserving MVV IN multimerization and intasome assembly. The side chains of Arg231 residues from various IN chains are directly involved in interactions with viral and target DNA (Figs. 1c and 2a). Finally, MVV IN His12 is part of the invariant HHCC motif that coordinates a $Zn^{2+}$ ion, which is critical for NTD structural stability[40,41]. Within the MVV intasome, the NTDs from numerous IN chains contribute to the CIC and/or to IN-vDNA and IN-IN interactions (Fig. 1a, d and Supplementary Fig. 3b). Therefore, both R231E and H12N were expected to disrupt MVV intasome formation.

We first evaluated the ability of the MVV IN variants to carry out strand transfer in vitro, utilizing short double-stranded oligonucleotide mimics of pre-processed MVV U5 vDNA ends. When supercoiled plasmid is used as target, two types of strand transfer products can be separated by agarose gel electrophoresis. Full-site products result from concerted insertions of pairs of vDNA oligonucleotides, linearizing the target plasmid. Uncoupled strand transfer of a single vDNA oligonucleotide yields half-site products, which migrate in agarose gels similar to the open circular form of the plasmid (Supplementary Fig. 5a). In agreement with published results, WT MVV IN displayed robust strand transfer activity, generating full-site and half-site products in the presence of LEDGF/p75 (Supplementary Fig. 5b)[8]. All MVV IN mutants from our panel displayed profound defects in enzymatic activity, with only F223A, R231E, Y261E, Y261A, V263E, and QQQ211PGG INs generating detectable levels of strand transfer products (Supplementary Fig. 5b). Quantification of overall strand transfer levels using real-time quantitative PCR revealed 30-fold or greater defects in activity across the mutant panel (Fig. 3b).

**Mutations targeting CTD-CTD interfaces and the CCD-CTD linker perturb MVV IN multimerization and intasome assembly in vitro.** To study effects of the amino acid substitutions on self-association of MVV IN, we analyzed the proteins by size exclusion chromatography coupled with multi-angle laser light scattering (SEC-MALLS). MVV IN proteins, at 1, 2, 4, and 8 mg/mL, were separated by chromatography through a Superdex-200 column and the molar mass distribution was resolved by MALLS. Under these conditions, WT MVV IN was predominantly tetrameric at the lowest protein concentration and formed larger species, likely dimers of tetramers, at higher input concentrations (Fig. 3c), recapitulating our previous observations[8] as well as similar studies based on HIV-1 IN[42,43]. As expected, E154Q and R231E behaved similarly to WT protein (Fig. 3c). At all input concentrations, WT, E154Q, and R231E INs displayed average molecular masses between those of a tetramer and octamer. The NTD is critical for IN tetramerization[33,44] and, concordantly, H12N IN eluted as a mixture of monomers and dimers (Fig. 3c). The mutants with substitutions at the CTD-CTD interfaces and the CCD-CTD linker displayed considerable and significant multimerization defects, displaying average molar masses below that of the IN tetramer at input concentrations of 2 and 1 mg/mL (Fig. 3c). This was particularly apparent with the linker variants QQQ211PGG, QQ207GP, and QQQ211PFV$_{304-314}$, which were predominantly dimeric, even up to 8 mg/mL.

We reasoned that the reduced ability of MVV IN mutants to assemble into tetramers and/or higher order multimers may affect their abilities to form intasomes and thus explain the observed defects in strand transfer activity (Supplementary Figs. 3b and 5b). To test this hypothesis, we monitored MVV IN assembly into intasomes in the presence of vDNA by size exclusion chromatography. As expected, WT protein and active site mutant E154Q efficiently formed high-molecular weight nucleoprotein complexes, with elution volumes expected for the intasome (Fig. 3d and Supplementary Fig. 5c)[8]. While R231E, Y261A, Y261E, and QQQ211PGG INs assembled into intasomes with greatly reduced yields (Fig. 3d), the remaining mutants failed to do so (Supplementary Fig. 5d). Notably, aside from the E154Q active site mutant control, the IN mutants capable of forming detectable intasome complexes also retained detectable levels of strand transfer activity (Fig. 3b).

**Disruption of the IN CTD-CTD interfaces or the α-helical CCD-CTD linker abrogates MVV infectivity.** To further investigate the importance of the hexadecameric intasome assembly, the mutations were introduced into a single-cycle MVV virus. We utilized a four-component MVV-derived lentiviral vector system comprising a Gag-Pol packaging construct, a transfer vector harboring the firefly luciferase reporter gene under the control of an internal cytomegalovirus (CMV) promoter, and plasmids expressing MVV Rev and vesicular stomatitis glycoprotein G (VSV-G) proteins[45,46]. We introduced H12N, E154Q, F223A, Y261A, V263E, and QQQ211PGG mutations into the IN-coding region of the packaging construct and produced VSV-G-pseudotyped vector particles. Human embryonic kidney 293T (HEK293T) cells were infected with the virus preparations, which were normalized by virion-associated reverse transcriptase (RT) activity levels. Measurement of luciferase activity 7 day post-infection revealed 50–100-fold infectivity defects of the mutants. Notably, infectivities of the F223A, Y261A, V263E, and QQQ211PGG viruses were comparable to those of the E154Q or H12N controls (Fig. 3e).

We next evaluated biochemical and virological properties of the IN mutant viruses to assess the nature of the infectivity defects. Immunoblotting of viral lysates revealed that the viruses

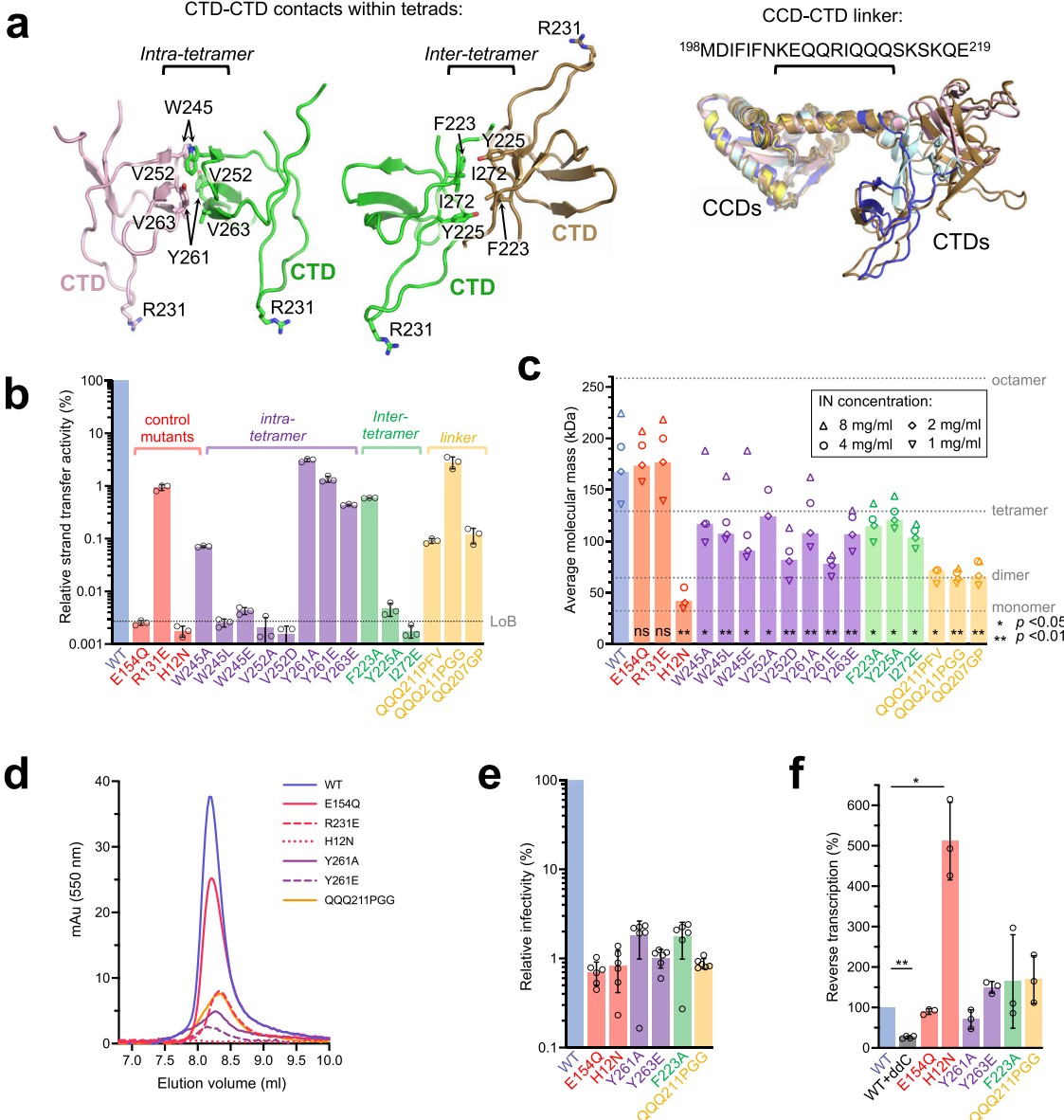

carrying single IN missense mutations harbored normal levels of mature p24 capsid protein (Supplementary Fig. 6a). Capsid levels were reduced approximately 3-fold in QQQ211PGG particles. However, this relatively minor maturation defect appeared incongruent with the ~100-fold defect in infectivity (Fig. 3e). Amino acid substitutions in HIV-1 IN often result in pleiotropic phenotypes, characterized by defects in vDNA synthesis during reverse transcription[47,48]. To determine whether the reduced luciferase levels reflected a defect occurring during integration or at an earlier step of MVV replication, we measured the levels of reverse transcription products in infected cells. As expected, WT and IN mutant reverse transcription in most cases peaked at 8 h post-infection, and WT vDNA synthesis was suppressed by 2′,3′-dideoxycytidine (ddC), a potent inhibitor of MVV RT[49] (Supplementary Fig. 6b). With the exception of H12N, which unexpectedly displayed a large increase of vDNA accumulation, the mutants did not display drastic perturbations at this stage of the viral life cycle (Fig. 3f and Supplementary Fig. 6b). The overcompensation of vDNA synthesis by the H12N IN mutant is in sharp contrast to the vDNA synthesis defects observed with H12N IN HIV-1[50,51]. While we cannot explain this difference, crucially, the MVV IN mutants targeting the CTD-CTD

interactions and the α-helical CCD-CTD linker within the MVV intasome did not display drastic reductions in vDNA levels, consistent with specific defects at the level of intasome assembly and the consequential loss of functional integration.

**Characterization of MVV intasome and LEDGF/p75 stoichiometry.** In accordance with previously determined crystal structures[32,33], the two copies of LEDGF/p75 within our STC reconstruction were bound at the IN CCD dimerization interface, making additional contacts with the IN NTD (Fig. 1d and Supplementary Fig. 7a). Harboring an IN hexadecamer, the MVV intasome presents sixteen potential LEDGF/p75 binding sites, all of which appear surface exposed and available for the interaction with the host factor (Supplementary Fig. 7b). As a result of the two-fold symmetry of the assembly, these sites form eight structurally non-equivalent position pairs. We reasoned that the pair of positions occupied by LEDGF/p75 in our STC structure may have the highest affinity for the host factor. To probe the ability of the MVV intasome to bind additional copies of LEDGF/ p75, we used single-molecule total internal reflection fluorescence (TIRF) microscopy. MVV intasomes were assembled using bio- tin- and Cy3-conjugated vDNA oligonucleotides and LEDGF/p75

**Fig. 3 Design and activities of MVV IN mutants. a** Locations of targeted IN residues of the CTD-CTD interfaces (left) and configuration of the CCD-CTDs linkers within eight structurally distinct IN chains of the intasome (right; amino acid sequence of the linker is shown above the superposition). Colors of protein chains are preserved from Fig. 1. Intra-tetramer and inter-tetramer CTD-CTD interfaces, the CCD-CTD linker, and specific amino acid residues targeted by mutagenesis (shown as sticks) are indicated. Due to C2 symmetry, the two CTD tetrads (Fig. 1b) are equivalent within the intasome. **b** Strand transfer activities of indicated MVV IN mutant proteins relative to that of WT (set to 100%), measured by real-time PCR. The bar plot displays mean values with standard deviations from n=3 independent experiments for each condition; open circles indicate values for individual repeat measurements. For clarity, bar plots are color-coded: blue bars show property of WT IN; red, control IN mutants; purple and green, mutants of the intra- and inter-tetramer CTD-CTD interfaces, respectively; yellow, mutants of the CCD-CTD linker. The gray dotted line represents the level of background (LoB), determined from three IN-omit reactions and defined as mean background +1 standard deviation. Qualitative analysis of the strand transfer products by agarose gel electrophoresis is shown in Supplementary Fig. 5b. **c** Average molar masses (kDa) of MVV IN variants determined by SEC-MALLS upon injections of the proteins at 8, 4, 2, and/or 1 mg/mL (indicated with upward triangles, circles, diamonds, and downward triangles, respectively). Bars represent values obtained with IN injected at 2 mg/mL. Molecular masses of MVV IN monomer (32.3 kDa), dimer, tetramer, and octamer are indicated with gray dotted lines. Statistical significance (WT vs mutant) was estimated using two-tailed paired Student's t-test, and the results are reported as highly significant (**$p < 0.01$), significant (*$p < 0.05$), or non-significant (n.s.); exact P values are provided in Source Data file. **d** Size exclusion chromatography elution profiles of CSC intasomes assembled with Cy3-labeled vDNA and WT or mutant MVV INs. The curves report Cy3 absorbance at 550 nm to distinguish nucleoprotein complexes from protein aggregates. Only elution volumes 7–10 mL are shown here; the complete elution profiles (0–20 mL), including results of the intasome assemblies with the remaining MVV IN mutants, are shown in Supplementary Fig. 5d. **e** Infectivity of single-cycle MVV-derived vectors produced using Gag-Pol constructs incorporating WT or indicated mutant IN. Luciferase expression was measured 7 day post-infection. The bars indicate mean values with standard deviations from $n = 6$ biological replicates for each condition; open circles indicate values for individual measurements. **f** Quantification of late reverse transcription products in cells infected with WT or IN mutant MVV vectors at 8 h post-infection. The bars show mean values with standard deviations from $n = 3$ biological replicates for each condition; open circles indicate values for individual measurements; two-tailed paired Student's t-test was used to estimate WT-vs-mutant statistical significance (**$p = 2 \times 10^{-5}$; *$p = 0.02$). Source data are provided as a Source Data file.

labeled with SNAP-Surface649 (Surf649) fluorophore. We purified the intasomes by size-exclusion chromatography to remove free vDNA, captured the nucleoprotein complexes in a streptavidin-coated microfluidic flow cell, and then incubated the immobilized intasomes with LEDGF/p75-Surf649. Individual spots containing Cy3 and Surf649 fluorescent signals were detected, and the decrease of Surf649 fluorescence intensity was monitored during photobleaching under conditions of variable ionic strength (Fig. 4a, b and Supplementary Movie 1). The results revealed an average of 6, 4, and 2 LEDGF/p75 subunits bound per MVV intasome in the presence of 0.2, 0.5, and 1 M NaCl, respectively, with up to 16 LEDGF/p75 subunits bound per intasome at the lowest NaCl concentration (Fig. 4c and Supplementary Table 2).

**LEDGF/p75 strongly influences MVV integration site selection.** LEDGF/p75 strongly stimulates in vitro strand transfer activity of MVV IN and other lentiviral INs[8,17,19,20]. To test if the presence of LEDGF/p75 or its paralog HRP2[30] in target cells influences MVV infectivity, we used HEK293T-derived cell clones ablated for one or both of these host factors. LKO is an LEDGF-null cell line, generated via TALEN-mediated *PSIP1* gene disruption[52]. In addition, we established a dual knockout clone, LHKO, which additionally lacked HRP2 due to disruption of *HDGFL2* gene (Supplementary Fig. 8a). As above, we infected these cells with RT-normalized quantities of WT or IN active site mutant E154Q MVV vectors. Measured 7 day post-infection, the infectivity of the WT vector in LKO and LHKO cells was reduced ~5-fold compared to that in parental HEK293T cells (Supplementary Fig. 8a). The infectivity of the E154Q IN active site mutant virus in HEK293T cells was ~2% of the WT vector, with the residual luciferase expression likely explained by persistence of non-integrated vDNA forms. However, WT MVV vector infectivity was not restored upon ectopic expression of ovine LEDGF/p75 in LHKO cells (Supplementary Fig. 8b). Hence, the lack of the host factor seemed unlikely to explain the observed reductions in MVV infection in these cells; instead, the clonal nature of LKO and LHKO cells may account for the reduced transduction efficiency. We accordingly next established a panel of ovine cell lines depleted of HRP2 and/or LEDGF/p75 via

synthetic RNA-directed CRISPR-Cas9 genome modification (Supplementary Fig. 9a). The parental cell line, CPT3, was derived from sheep choroid plexus cells immortalized by co-expression of simian virus 40 large T antigen and human telomerase[53]. Importantly, the resulting knockout cell lines (CPT3-LKO1, 2, 3, and 4, and CPT3-LHKO1 and 2) did not undergo single cell cloning. Transduction of the knockout cell panel with the MVV vector did not reveal consistent infectivity defects associated with LEDGF/HRP2 knockout status (Supplementary Fig. 9b), indicating that the host factors are not essential for MVV infection.

LEDGF/p75 and, to a lesser degree, its paralog HRP2 direct HIV-1 integration towards bodies of highly expressed genes[22,25,26,35,54]. To test if LEDGF/p75 plays a similar role in guiding MVV integration, we mapped MVV vector integration sites in our panel of knockout cell lines. Human HEK293T, LKO, LHKO cells and ovine CPT3, CPT3-LKO1, CPT3-LHKO1, and CPT3-LHKO2 cells were infected with the WT MVV vector, and genomic DNA was isolated 5 day post-infection. Chromosomal junctions at integrated U5 vDNA ends were amplified using linker-mediated PCR, sequenced using Illumina technology and aligned with human and sheep genomes. We then correlated the distributions of uniquely mapped MVV integration sites with positions of annotated genes and their transcriptional activity, locations of transcription start sites (TSSs), CpG islands as well as with local A/T content and gene density (Tables 1 and 2). In addition, as the locations of human constitutive lamina-associated regions (cLADs, corresponding to perinuclear heterochromatin) and nuclear speckle-associated domains (SPADs, corresponding to transcriptionally-active regions) were available[55,56], these features were also used in the analyses of MVV integration sites in human cells (Table 1).

These analyses revealed that MVV displayed a strong bias towards integration into gene bodies in both human (HEK293T) and ovine (CPT3) cell lines (Tables 1 and 2; results of statistical tests are given in Supplementary Tables 3 and 4). In total, 72.1% of MVV integration sites were found within annotated transcription units in the sheep genome, compared to the expected frequency of 43.3% determined using an in silico-generated matched random control (MRC). Similar to HIV-1 and other

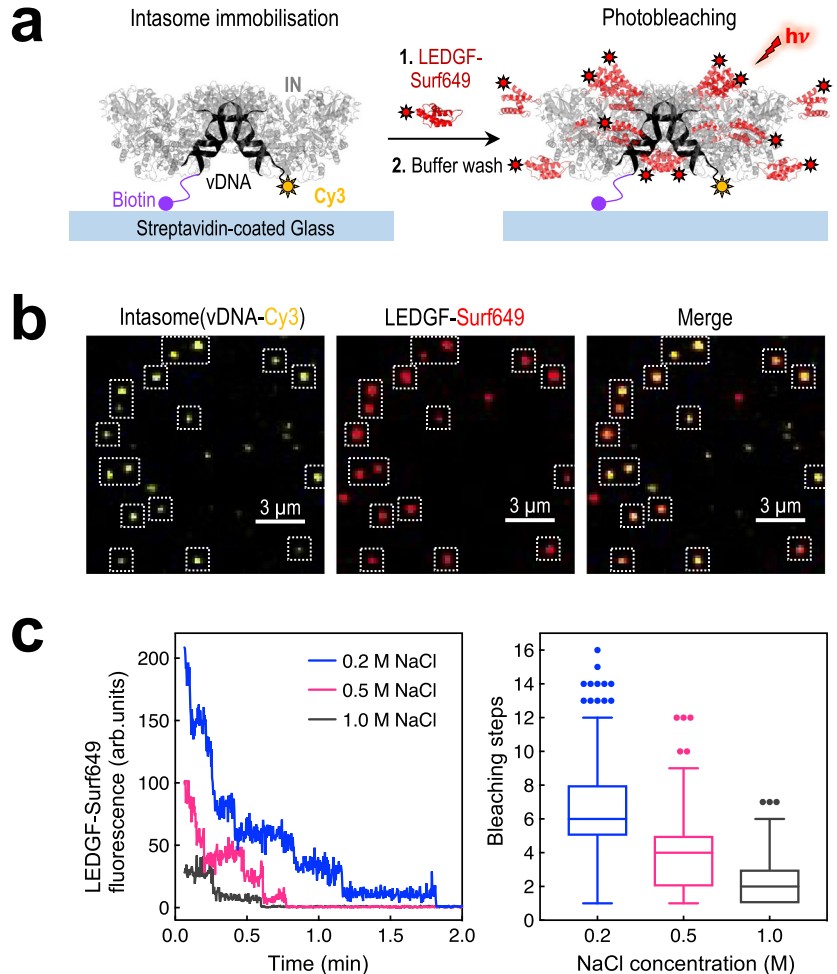

**Fig. 4 Quantitation of intasome-LEDGF/p75 stoichiometry by single-molecule TIRF microscopy. a** Schematic of the photobleaching experiment. Intasomes containing biotin- and Cy3-conjugated vDNA were immobilized on a streptavidin-coated cover slip (light blue rectangle). IN and vDNA are shown as gray and black cartoons, respectively; biotin and Cy3 are depicted as a purple circle and yellow solar symbol, respectively. Following incubation with LEDGF/p75 (red cartoons) conjugated with Surf649 (red solar symbol) and a wash with buffer containing 0.2, 0.5, or 1 M NaCl, immobilized intasomes were observed by TIRF microscopy. The individual steps of Surf649 photobleaching during illumination with a 640-nm laser (h$\nu$) were counted. **b** Representative images of surface attached intasome (vDNA-Cy3, yellow; LEDGF/p75-Surf649, red) molecules. The field of view of 14.4 by 14.4 μm shown here is representative of the dataset, which included three areas of 81.9 by 81.9 μm (see Methods section for details). Dotted, white line squares indicate individual intasome-LEDGF/p75 complexes. **c** Left: examples of stepwise photobleaching traces of LEDGF/p75-Surf649 at increasing NaCl concentration: 0.2 M (blue), 0.5 M (magenta), and 1.0 M (dark gray). The vertical axis represents fluorescence in arbitrary units (arb. units). Right: Box-and-whiskers plots summarizing statistical analysis of the number of LEDGF/p75-Surf649 photobleaching steps per intasome, for various NaCl concentrations (see Supplementary Table 2 for further details). Each box encloses data between 25th and 75th percentiles, with the median value displayed as a horizontal line. Vertical lines (whiskers) indicate 10th and 90th percentiles; outliers are indicated with closed circles. Source data are provided as a Source Data file.

**Table 1 HIV-1 and MVV integration site distributions in human HEK293T, LKO, and LHKO cells.**

| Virus or control[a] | Cells | Unique sites | RefSeq genes (%)[b] | ±2.5 kb TSS (%)[c] | ±2.5 kb CpG (%)[c] | cLADs (%)[b] | SPADs (%)[b] | Gene density[d] |
|---|---|---|---|---|---|---|---|---|
| MRC | in silico | 265,399 | 45.0 | 3.8 | 3.8 | 24.5 | 3.7 | 8.5 |
| HIV-1[e] | HEK293T | 43,230 | 83.3 | 4.2 | 5.5 | 5.7 | 29.8 | 20.9 |
| HIV-1[e] | LKO | 28,177 | 63.1 | 10.2 | 11.6 | 10.7 | 15.3 | 14.4 |
| MVV | HEK293T | 411,721 | 67.0 | 2.9 | 2.6 | 19.2 | 3.8 | 9.1 |
| MVV | LKO | 2235 | 47.7 | 7.5 | 7.6 | 23.3 | 8.0 | 10.3 |
| MVV | LHKO | 24,321 | 49.1 | 7.8 | 8.6 | 23.9 | 7.6 | 9.9 |

[a]Matched random control (MRC) or a single-cycle GFP vector (HIV-1 or MVV).
[b]Integration events inside a genomic feature (RefSeq gene, cLAD, or SPAD).
[c]Integration events within 2.5 kb of a feature (TSS or CpG island).
[d]Average number of RefSeq genes within a 1-Mb window (±500 kb) of integration site.
[e]HIV-1 integration sites were from refs. 58,113.

**Table 2 MVV integration site distributions in ovine CPT3 cells in the presence and absence of HRP2 and/or LEDGF/p75.**

| Virus or control[a] | Cells | Unique sites | RefSeq genes (%)[b] | ±2.5 kb TSS (%)[c] | ±2.5 kb CpG (%)[c] | Gene density[d] |
|---|---|---|---|---|---|---|
| MRC | in silico | 284,737 | 43.3 | 4.6 | 6.1 | 19.6 |
| MVV | CPT3 | 296,801 | 72.1 | 5.2 | 4.4 | 26.8 |
| MVV | CPT3-LKO1 | 145,863 | 51.8 | 12.4 | 16.5 | 29.1 |
| MVV | CPT3-LHKO1 | 3880 | 47.6 | 10.9 | 14.9 | 27.9 |
| MVV | CPT3-LHKO2 | 82,570 | 51.6 | 12.4 | 16.0 | 28.2 |

[a]Matched random control (MRC) or a single-cycle MVV GFP vector.
[b]Integration events inside a genomic feature (RefSeq gene, intergenic region).
[c]Integration events within 2.5 kb of a feature (TSS or CpG island).
[d]Average number of RefSeq genes within a 1-Mb window (±500 kb) of integration site.

lentiviruses[22,25,57], the frequency of MVV integration into genes significantly decreased in LEDGF-null human ($p \sim 10^{-77}$) and ovine ($p < 10^{-300}$) cells, although we did not reproducibly observe an additional defect upon HRP2 ablation. The frequency of MVV genic integration events strongly correlated with transcription activity. Thus, genes expressed at the highest level were ~4-fold more likely to host an MVV integration event than those expressed at the lowest level; this correlation decreased significantly in the absence of LEDGF/p75 in both human (Fig. 5a) and ovine (Fig. 5b) cells. Similar to HIV-1, the frequency of MVV integration in proximity of TSSs and CpG islands significantly increased in LEDGF-null cells. LEDGF/p75 contains an AT-hook, which was implicated in DNA binding[29,58]. Concordantly, LEDGF/p75 depletion shifted HIV-1[22,25] (Fig. 5c) and MVV (Fig. 5c, d) integration events towards regions with lower A/T content. In accordance with the cryo-EM structure (Fig. 2a), ablation of HRP2 and/or LEDGF/p75 did not affect local nucleotide preferences at the integration sites (Supplementary Fig. 9c), which remained basically unchanged from those observed with in vitro-assembled MVV intasomes (Fig. 2b). By sharp contrast to HIV-1, MVV did not display a strong preference for gene-dense regions or SPADs, and only moderately avoided integration into cLAD perinuclear chromatin. In HEK293T cells, HIV-1 integrated into SPADs ~8-fold more frequently than expected, and this preference was reduced ~2-fold upon LEDGF/p75 ablation (Table 1). By contrast, the frequency of MVV integration into SPADs nearly precisely matched the expected value of 3.7% and increased 2-fold in LKO ($p \sim 4 \times 10^{-19}$) and LHKO cells ($p \sim 10^{-149}$). Although we have not studied the effect of HRP2 ablation in isolation, distributions of MVV integration sites in double knockout LHKO and CPT3-LHKO cells tended to mirror those observed in LKO and CPT3-LKO, respectively.

## Discussion

Our cryo-EM reconstructions provide snapshots of a lentiviral intasome in its two key functional states: prior to capture of tDNA and following strand transfer. In agreement with the previous low-resolution structure[8], the MVV intasome contains sixteen IN subunits in a tetramer-of-tetramers arrangement. By comparison, in vitro assembly of HIV-1 and SIV$_{rcm}$ intasomes led to formation of highly heterogenous nucleoprotein complexes containing from 4 to 16 IN subunits, with a prominence of dodecamers, as well as linear stacks of the intasomes[13,17,18]. These observations underscore the utility of the MVV intasome as a convenient model to study lentiviral integration and call for further systematic studies of the lentiviral intasome architecture and its functional significance.

Notwithstanding their heterogeneity, the primate lentiviral intasomes are closely related to their MVV counterpart (see Cook et al.[17] for details). Importantly, the lentiviral intasomes share two unique features: intra-tetramer CTD-CTD contacts and

an α-helical configuration of the CCD-CTD linkers (Fig. 3a). These features do not participate in the formation of the CIC, which is the functional core of the retroviral intasome. The MVV intasome also employs intra-tetramer CTD-CTD interactions within CTD tetrads (Figs. 1b and 3a). In solution, MVV IN exists as tetramers, which can self-associate into larger multimers (Fig. 3c)[8]. Predictably, disruption of the CTD-CTD interfaces or the CCD-CTD α-helical linker significantly reduced the multimerization state of the protein (Fig. 3c). Moreover, the mutations rendered MVV IN unable to catalyze strand transfer, assemble into intasomes in vitro and caused gross defects in the context of a single-round infection (Fig. 3). These results argue that MVV IN evolved to function in a highly multimeric form and is non-functional when forced into smaller sub-complexes. Because HIV-1 particles package ~100 Gag-Pol molecules, the number of available IN subunits is unlikely to limit intasome assembly during lentivirus infection[59,60]. Moreover, given the proclivity of lentiviral INs to multimerize (Fig. 3c)[19,33,61], we cannot rule out that lentiviral pre-integration complexes may contain even larger IN assemblies. While ascertaining the true multimeric state of lentiviral IN during integration may require visualization of viral nucleoprotein complexes in infected cells, our results argue that the functional intasome must assemble from multiple IN tetramers.

A common property of lentiviral INs is the interaction with LEDGF/p75[20,23], which was shown to direct HIV-1 integration towards active TUs[22,25]. Similarly, LEDGF/p75 regulates integration of non-primate lentiviruses such as feline immunodeficiency and equine infectious anemia viruses[57,62], and, as we have shown here, plays a dominant role to direct MVV integration (Fig. 5). The interaction of the MVV intasome with LEDGF/p75 is sensitive to buffer conditions (Fig. 4c), and only two IBDs could be located in our STC cryo-EM reconstruction (Supplementary Figs. 2a and 3c). Using single-molecule approaches, we showed that the intasome can bind many additional LEDGF/p75 molecules (Fig. 4c). This finding is consistent with the availability of 16 LEDGF/p75 binding sites on the IN hexadecamer (Supplementary Fig. 7b), which may display different affinities for the host factor. The IN interface with LEDGF/p75 is bipartite, involving a IN CCD dimer and associated NTD[33,63]. Indeed, the affinity of the HIV-1 IN interaction with LEDGF/p75 in the absence of the NTD was greatly reduced[64]. The two LEDGF/p75 IBDs resolved in our structure are engaged by the domain-swapped NTDs involved in the synaptic interface, and are thus stabilized within the structure via contacts with vDNA (see blue and green IN chains in Fig. 1a, c, d and Supplementary Figs. 3c and 7a). The remaining 14 NTDs of the IN hexadecamer are more loosely associated with the body of the intasome, which may reduce the affinity of the corresponding sites for LEDGF/p75.

We propose that the ability of the lentiviral intasome to bind a large number of LEDGF/p75 molecules (Fig. 4c and Supplementary Fig. 7b) allows the PIC to establish highly multivalent

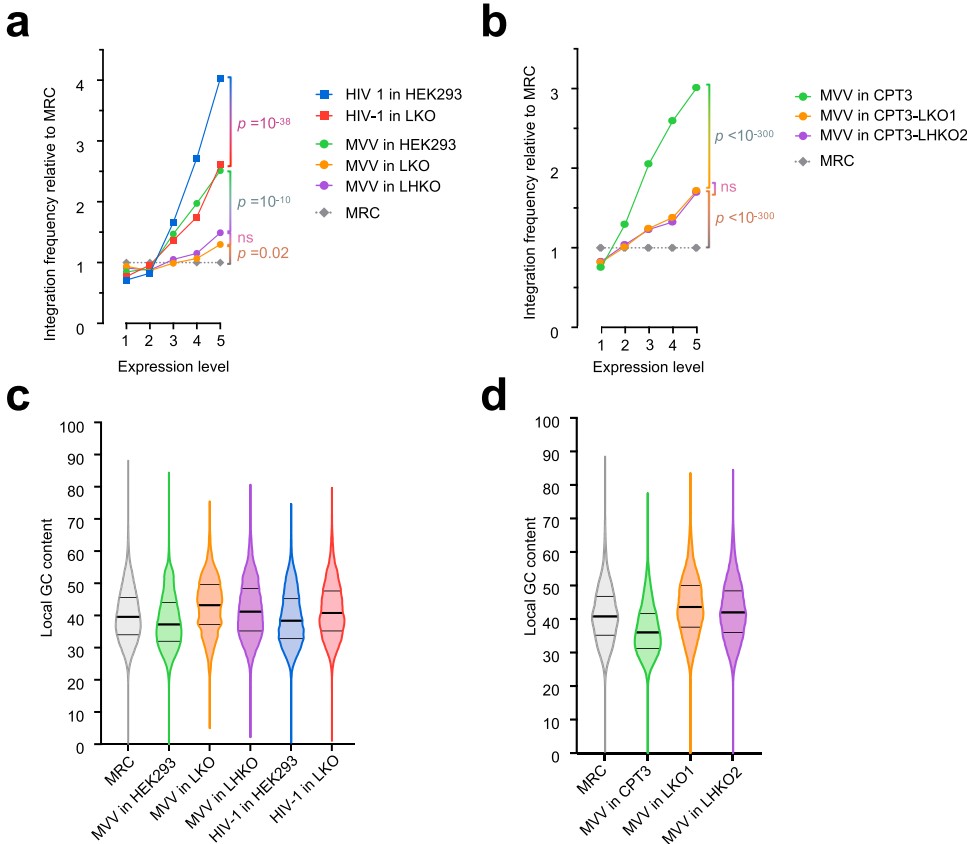

**Fig. 5 Effects of LEDGF/p75 depletion on MVV integration site distribution. a** Frequencies of MVV and HIV-1 integration events in HEK293T cells (green circles and blue squares, respectively), in LKO cells (red squares and orange circles, respectively), MVV integration events in LHKO cells (purple circles), and matched random control (MRC) sites (gray diamonds) into TUs of variable transcriptional activity. Human TUs were ranked by their activity into five bins, where each bin contained the same fraction of the genome; only integration events mapped to RefSeq genes were considered for this analysis. Statistical significance (e.g. HIV-1 in HEK293T vs LKO, as indicated by vertical brackets) was determined using two-sided Chi-squared tests for corresponding 2 × 5 contingency tables. All data are provided as a Source Data file. **b** Frequency of MVV integration into TUs of different activity in ovine CPT3 (green circles), CPT3-LKO1 (orange circles), and CPT3-LHKO2 cells (purple circles). **c, d** Local GC contents for mapped integration sites in human (**c**) and ovine (**d**) cells. The data are plotted as violin plots showing frequency distribution for individual GC contents. Thick horizontal lines represent median values, while thin lines indicate boundaries of 25th and 75th percentiles of data points. See Supplementary Tables 3 and 4 for statistical analyses of panel **c, d** datasets. Source data are provided as a Source Data file.

interactions with chromatin. Simultaneous contacts with multiple H3K36Me3-containing nucleosomes can only form in locations enriched for the epigenetic mark. Such a mechanism may allow lentiviral PICs to discriminate H3K36Me3 peaks on the backdrop of noisy epigenetic landscapes, providing a possible biological rationale for the expansion of the lentiviral intasome. Interestingly, similar to other non-primate lentiviruses[57], MVV does not show preference for integration into SPADs (Table 1). This observation is consistent with the idea that recruitment of CPSF6, the factor responsible for HIV-1 targeting to nuclear speckles, is a recent acquisition, specific to primate lentiviruses[57].

Notwithstanding the profound effects on lentiviral integration site distribution, ablation of LEDGF/p75 results in highly variable and often modest defects in single-round HIV-1 infection (Supplementary Fig. 9b)[22–25,57,62,65,66]. Similarly, redirection of gamma-retroviral integration via mutations disrupting IN binding to cellular bromodomain and extra-terminal domain proteins revealed little effect on viral infectivity[67]. Even more strikingly, despite redirecting a considerable fraction of proviruses to perinuclear heterochromatin, ablation of CPSF6 slightly increases infectivity of HIV-1 and other primate lentiviruses in single-cycle assays[57]. Modest infectivity defects were also observed in spumaretroviruses where the Gag mutant R540Q quantitatively redirected PFV integration into centrosomes[68]. Therefore, given

these observations, we conclude that single-cycle infectivity assays do not replicate the conditions that led to the emergence of the observed integration preference traits. Integration site selection is likely to confer subtle transcriptional advantages that nonetheless translate to important growth advantages within the infected host, e.g., to outpace virus-induced cytopathicity and/or humoral immune pressure[69].

Our structures highlight general features of the retroviral integration machinery and reveal intriguing differences between viral genera. Notably, as in the case of PFV, MVV strand transfer appears to result in ejection of the phosphodiester bond linking viral and target DNA from the IN active site, suggesting conservation of the mechanism proposed to discourage the reversal of the integration process[4,9]. Indeed, a much more distantly related Mu phage transposase was also shown to utilize this mechanism[70]. The ejection of the transesterified phosphodiester group is likely driven by tension caused by sharp deformation of the tDNA duplex. The size of the tDNA segment separating insertion sites of the two vDNA ends into opposing tDNA strands ranges between 4 and 6 bp, depending on the retroviral species. By contrast, the distance between intasomal active sites remains relatively constant between retroviral intasomes (28–30 Å, defined as the distance between Cα atoms of the invariant IN active site Glu residues). This explains the marked differences in

tDNA conformation induced by diverse retroviral intasomes (Supplementary Fig. 4). Evidently, in each case, the observed deformation allows sufficient expansion of the tDNA major groove to afford intasomal active sites access to the scissile phosphodiester bonds. The propensity of retroviruses and retrotransposons to integrate into nucleosomes is supported by strong evidence[71–73], and the PFV intasome in complex with a core nucleosome particle was visualized by cryo-EM[74,75]. However, wrapping around the histone octamer imposes constraints of the conformation and major groove accessibility of nucleosomal DNA[76]. Thus, the differences in tDNA conformation during its capture by the retroviral intasomes strongly support the idea that retroviruses may have evolved genus or even species-specific ways of interacting with chromatinized target DNA[77].

## Methods

**Recombinant protein expression and purification**. The plasmid pCPH6P-MVV-IN, encoding IN from the KV1772 MVV isolate[78] with a cleavable hexahistidine tag, was previously described[8]. Mutations were introduced into pCPH6P-MVV-IN using PCR. Plasmid pET22b-MVV-CAfl, encoding the MVV capsid/p24 protein, was constructed by Gibson assembly as recommended by the manufacturer (New England Biolabs) using PCR fragment amplified from pCAG-MV-GagPol-IN$^{KV1772}$-CTEx2[45,46] and *Nde*I/*Xho*I-linearized pET-22b(+) (Millipore Sigma). MVV IN proteins and human LEDGF/p75 were produced in bacteria and purified as previously described[8]. The construct used for expression of LEGDF-SNAPf, which was used in TIRF microscopy, was made by ligating a PCR fragment encoding full-length human protein between *Apa*I and *Bam*HI sites of pHis$_{10}$-PS-SNAPf[79] (gift from Ron Vale; Addgene plasmid #78512; http://n2t.net/addgene:78512; RRID:Addgene_78512). LEGDF-SNAPf protein was labeled with SNAP-Surface649 (Surf649, New England Biolabs). Briefly, 8 µM LEDGF/p75-SNAPf was incubated with 16 µM Surf649 in 500 mM NaCl, 5 mM DTT, 50 mM Tris-HCl, pH 7.5 for 30 min at 37 °C. Labeled protein was purified from excess dye using a Bio-Spin P6 column (BioRad); labelling was monitored by native protein mass spectrometry (Mass Spectrometry Facility, University of St. Andrews, UK).

Recombinant MVV capsid/p24 protein was produced in *E. coli* BL21(DE3) cells transformed with pET22b-MVV-CAfl. Cells were grown in shaker flasks at 37 °C to an OD$_{600}$ of 0.6, and protein expression was induced with 1 mM isopropyl β-D-1-thiogalactopyranoside for 4 h at 37 °C. Pelleted cells resuspended in lysis buffer (50 mM Tris-Cl pH 8.0, 50 mM NaCl, 20 mM imidazole, 1 mM TCEP) were flash-frozen. Thawed resuspensions were lysed by sonication for 2 min on ice in 5 s intervals, with 30 s gaps between bursts. The supernatant fraction following centrifugation at 50,000 g for 30 min was loaded onto a HisTrap HP column (GE Healthcare). After extensive washing with lysis buffer, the column was developed with elution buffer (50 mM Tris-Cl pH 8.0, 50 mM NaCl, 250 mM imidazole, 1 mM TCEP). Eluted protein, concentrated by ultrafiltration, was loaded onto a Superdex S200 10/300 gel-filtration column equilibrated in SEC buffer (50 mM Tris-Cl pH 8.0, 50 mM NaCl, 1 mM TCEP). Pooled fractions of capsid protein were flash-frozen in liquid nitrogen and stored at −80 °C. For antibody production, thawed protein was dialyzed against phosphate-buffered saline (PBS) overnight at 4 °C. The protein was used for production and affinity purification of polyclonal rabbit anti-CA/p24 antibody (Thermo Fischer Scientific).

**MVV intasome assembly and purification**. HPLC-purified oligonucleotides were purchased from Sigma-Aldrich. The MVV CSC intasome complex was assembled using a double-stranded oligonucleotide mimicking the terminal 29 bp of the processed MVV U5 vDNA end prepared by annealing synthetic oligonucleotides EV272 (5′-CCGTGCAACACCGGAGCGGATCTCGCA) and EV273 (5′- GCTG CGAGATCCGCTCCGGTGTTGCACGG). When required, EV272 was modified with 5′-Cy3 dye (for TIRF microscopy, photobleaching and quantitative intasome assembly assays) and EV273 with 3′- triethylene glycol (TEG) biotin (in TIRF microscopy and photobleaching experiments). The MVV STC was assembled using a DNA construct corresponding to the product of full-site integration of 23-bp MVV U5 viral DNA end (vDNA) mimic into a palindromic 52-bp target DNA. The branched DNA was made by annealing oligonucleotides 5′-GCTGCGAGA TCCGCTCCGGTGTT, 5′-AACACCGGAGCGGATCTCGCAGCC ACC CTAATCAAGTTTTTTGGGG and 5′-CCCCAAAAAACTTGATTAGGGTG; the palindromic tDNA portion (underlined) was designed based on the preferred integration site sequence in pGEM9zf plasmid[8]. The intasomes were typically assembled by incubating 7 µM MVV IN, 8 µM LEDGF/p75, and 3.75 µM annealed vDNA (or the strand transfer product mimic) in 80 mM NaCl, 40 mM potassium acetate, 3 mM CaCl$_2$, 10 µM ZnCl$_2$, 1 mM dithiothreitol (DTT) and 25 mM Bis-Tris-HCl, pH 6.0 in a total volume of 200 µl at 37 °C for 10 min. To prepare samples for cryo-EM, the reaction was upscaled to a total volume of 1 mL. The opalescent mixture was supplemented with 50 mM BisTris-HCl, pH 6.5 and 190 mM NaCl and incubated on ice for 5 min to clear. If the starting volume exceeded 200 µL, the mixture was concentrated by ultrafiltration in a VivaSpin device to a final volume of 200 µL. Intasomes were purified by size exclusion

chromatography on a Superdex-200 10/30 column (GE Healthcare) pre-equilibrated in 310 mM NaCl, 3 mM CaCl$_2$, 25 mM BisTris-HCl, pH 6.5. Chromatograms were recorded on an ÄKTA Purifier system using Unicorn version 5.31 software (GE Healthcare).

**MVV CSC cryo-EM grid preparation, data collection, and image processing**. MVV CSC intasomes, assembled and purified as previously described[8], were applied onto R1.2/1.3 gold UltrAufoil grids, Au 300 mesh (Quantifoil). Cryo-EM grids were prepared by manually freezing using a manual plunger in a cold room at 4 °C and stored in liquid nitrogen for future data acquisition. Cryo-EM movie frames were collected on a Talos Arctica transmission electron microscope (Thermo Fisher Scientific) operating at 200 keV equipped with a K2 summit direct detector (Gatan). Data collection was performed using the Leginon software version 3.0[80,81] at a magnification of 45,000x, corresponding to a pixel size of 0.92 Å/pixel in nanoprobe mode. The stage was tilted to 40° during data collection to account for the preferential orientation of the sample within the vitreous ice[82]. Movies composed of 100 frames were collected in counting mode over 10 s (100 ms per frame). The total fluence was 43.6 e$^-$/Å$^2$ at a rate of 3.7 e$^-$/pix/s. Imaging parameters are summarized in Supplementary Table 1.

The movie frames were motion-corrected and dose-weighted using MotionCor2 version 1.4.0[83] on 6 by 6 patch squares and using a B-factor of 100 Å$^2$. The gain reference used for MotionCor2 was generated by using the Sum_all_tifs program, which is distributed within the cisTEM image processing suit[84]. The motion-corrected micrographs were imported into cryoSPARC version 3.2.0[85], which was then used to perform patch CTF estimation and particle selection. Manually selected particles were initially extracted with a box size of 384 pixels and then used to perform 2D classification. The class averages with different views from this initial round of 2D classification were used as 2D templates for template-based particle selection in cryoSPARC, and then the selected particles were extracted and subjected to another round of 2D classification to produce improved 2D classes characterized by slightly different views. We performed iterative cycles of template-based particle selection and 2D classification until we were able to maximize the recovery of different views of particles. For template picking, we set a particle diameter of 180 Å with an overlap that did not allow any two picks to be closer than 0.4 units of particle diameter in distance. 926,176 particles were extracted with a box size of 384 pixels, following particle inspection. This stack was then subjected to downstream processing. We performed several rounds of 2D classification with this stack, leaving 466,246 particles from best 2D classes. The remaining particles were then subjected to heterogeneous refinement using an imported map of the MVV CSC intasome (EMDB-4138)[8] in cryoSPARC. The particles were split into two classes through heterogeneous refinement, and the best class (based on visual inspection and measurement of resolution) was selected to perform one round of non-uniform refinement[86]. The refined map from non-uniform refinement was used as an input for the next round of heterogeneous refinement to further sort out bad particles. This iterative process, consisting of heterogeneous refinement and non-uniform refinement, was continued until we observed no further improvements to the resolution, as measured by the Fourier shell correlation (FSC) between reconstructions from pairs of independent half-sets[87] (Supplementary Fig. 10c). At this point, 147,860 particles remained and were used for the final non-uniform refinement, which was also combined with per-particle defocus refinement. The final global resolution was calculated in cryoSPARC as 3.43 Å using FSC analysis with a fixed threshold at 0.143 (Supplementary Fig. 1a). The local resolution was calculated using cryoSPARC (Supplementary Fig. 2a). The 3D FSC was obtained by using the 3D FSC server (https://3dfsc.salk.edu)[82] and the sampling compensation factor (SCF) and surface sampling plots were calculated using the graphical user interface tool (https://www.github.com/LyumkisLab/SamplingGui)[88,89]. Selected image analysis results are shown in Supplementary Fig. 10 and summarized in Supplementary Table 1.

**MVV STC cryo-EM grid preparation, data collection, and image processing**. A 4-µL drop of freshly prepared MVV STC at 0.15 mg/mL in 310 mM NaCl, 3 mM CaCl$_2$ and 25 mM BisTris-HCl, pH 6.5 was applied onto glow-discharged lacey carbon grids coated with 3-nm carbon film (Ted Pella, product code #01824). The grids were incubated for 60 s under 100% humidity in a Vitrobot Mark IV (FEI) at 20 °C. To reduce salt concentration, the grids were blotted for 0.5 s, immediately re-hydrated with a 4-µL drop of 75 mM NaCl and 25 mM BisTris-HCl, pH 6.5 and blotted again for 3.5 s, followed by plunging into liquid ethane. Cryo-EM data were collected on a Titan Krios electron microscope (FEI) operating at 300 keV equipped with a K2 Summit direct electron detector (Gatan) using EPU software version 1.9 (FEI). A total of 12,679 micrograph movie stacks were acquired at a magnification of 36,232, resulting in a pixel size of 1.38 Å at the specimen level, using a total fluence of 50 e/Å$^2$ spread over 50 frames with a defocus range of −1.0 to −4.5 µm.

The movies were corrected for drift and beam-induced motion applying dose weighting as implemented in MotionCor2 version 1.4.0[83]. The contrast transfer function was estimated using CTFFIND4 version 4.1.5[90] via Relion-2.1 interface[91]. Resulting frame sums were examined individually, and those containing mostly lacey carbon or showing evidence of crystalline ice contamination were discarded. A remaining 11,760 micrographs were used for further image processing (Supplementary Fig. 11a). Particles, picked manually on a subset of micrographs

using EMAN Boxer from Eman2 version 2.07[92], were subjected to reference-free 2D classification. Six well-defined 2D class averages, low-pass filtered to 20 Å, were used as references for automated particle picking in Relion-2.1 of the entire dataset. Particles picked along carbon edges were removed using Eraser tool in EMAN Boxer. The remaining 2,022,321 particles were extracted, with 2-fold binning applied, and subjected to several rounds of reference-free 2D classification in Relion-2.1. A total of 684,262 particles belonging to well-resolved 2D classes (Supplementary Fig. 11b) were subjected to 3D classification in Relion-2.0 into four classes without imposing symmetry (Supplementary Fig. 11c); 220,096 particles belonging to the single high-resolution class were re-extracted without binning, with a box size of 320×320 pixels, and used for 3D refinement without imposing symmetry. The resulting reconstruction showed features suggesting presence of IBD bound to two symmetric positions within the CIC. To improve occupancy of LEDGF/p75, the particles were subjected to 3D classification without re-alignment with a mask focused on the IBDs and the DNA component of the STC. Prior to classification, all IN-derived signal was removed, using particle subtraction utility in Relion-2.1. The procedure allowed isolation of a 3D class harboring two IBDs and comprising 121,619 particles (Supplementary Fig. 11d). Upon reverting to original non-subtracted particles, this subset was subjected to Bayesian polishing in Relion-3.1. The final reconstruction was obtained using non-uniform refinement following local and global CTF refinement (per-particle defocus and beam tilt, respectively), as implemented in cryoSPARC version 3.2.0. To aid in model building and for illustration purposes, the map was filtered and sharpened using deepEMhancer[93] or using density modification procedure in Phenix[94]. Details of EM processing statistics are given in Supplementary Table 1. Resolution is reported according to the gold-standard FSC using the 0.143 criterion[95,96] (Supplementary Fig. 1b), and local resolution was calculated using cryoSPARC (Supplementary Fig. 2a). The 3D FSC was obtained by using the 3D FSC server (https://3dfsc.salk.edu)[82] and the sampling compensation factor (SCF) and surface sampling plots were calculated using the graphical user interface tool (https://www.github.com/LyumkisLab/SamplingGui)[88,89].

**Real-space refinement.** The STC model was constructed from the original low-resolution strand transfer complex model published previously (PDB code 5M0R)[8] and coordinates for LEDGF/p75 IBD (PDB code 3HPH)[33]. The coordinates were docked into the cryo-EM map using UCSF Chimera[97]. Adjustments were made to the model interactively using Coot version 0.9.8[98] and the coordinates were subjected to real-space refinement in Phenix dev-4213-000 employing C2 NCS constraints[99]. The final model has good fit to the cryo-EM map (CC = 0.77) and reasonable stereochemistry as assessed using MolProbity version 4.5[100]. The refined STC model, with the tDNA removed, was then docked into the CSC cryo-EM map using UCSF Chimera[97]. To address slight differences in the STC and CSC structures, individual domains that were not well fitted to the CSC map were docked as rigid bodies to achieve a best-fit starting model. Two (C2 symmetry-related) NTDs (IN residues 1-35, in chains D and L) were removed from the model due to lack of supporting map. Manual adjustments were made to the model using Coot[98] and the coordinates were subjected to real-space refinement in Phenix dev-4213-000 employing C2 NCS constraints[99]. The final model has good fit to the cryo-EM map (CC = 0.70) and reasonable stereochemistry as assessed using MolProbity[100]. Details of the map and model statistics are given in Supplementary Table 1. Model-vs-map FSCs for both structures are shown in Supplementary Fig. 1c. The final cryo-EM reconstructions were deposited with the EMDB under accession codes EMD-26322 (CSC) and EMD-14453 (STC) and fitted coordinates with the PDB under accession codes 7U32 (CSC) and 7Z1Z (STC).

**SEC-MALLS.** All measurements were conducted using a Jasco chromatography system equipped with a DAWN-HELEOS II laser photometer and an OPTILAB-TrEX differential refractometer (Wyatt Technology). MVV IN was diluted to 1, 2, 4, or 8 mg/mL in 1 M NaCl, 7.5 mM 3-[(3-cholamidopropyl) dimethylammonio]−1-propanesulfonate, 2 mM DTT, 25 mM BisTris propane-HCl, pH 7.0 prior to injection of 20 µL samples onto a Superdex 200 Increase 3.2/300 column (GE Healthcare) equilibrated in 1 M NaCl, 3 mM NaN₃, 25 mM BisTris-HCl, pH 6.5. Chromatography was performed at 20 °C and a flow rate of 150 µL/min. Weight-averaged molecular masses of eluting species were calculated using the data from both detectors in ASTRA-6.1 software (Wyatt Technology).

**In vitro MVV IN strand transfer assays.** Strand transfer assays were carried out as previously described[8] using double-stranded oligonucleotide mimicking the terminal 29 bp of the processed MVV U5 vDNA end (Supplementary Fig. 5). The vDNA substrate was prepared by annealing synthetic HPLC-purified oligonucleotides 5'-CCGTGCAACACCGGAGCGGATCTCGCA and 5'- GCTGCGAGATCCGCTCCGGTGTT GCACGG. Reactions were initiated by addition of 1.1 µM MVV IN to 1.5 µM LEDGF/p75, 0.75 µM vDNA substrate and 7.5 ng/µL super-coiled target pGEM9zf DNA (Promega) in 25 mM BisTris-HCl, pH 6.0, 40 mM KCl, 5 mM MgSO₄, 5 µM ZnCl₂ and 2.5 mM DTT, in a final volume of 40 µL. Reactions were allowed to proceed for 1 h at 37 °C and were stopped by addition of 25 mM EDTA and 0.5% SDS. DNA products, deproteinized by digestion with proteinase K (Thermo Fisher Scientific) and ethanol precipitation, were dissolved in 20 µL of deionized water. Products were analyzed by electrophoresis through

1.5% agarose gels in Tris-Acetate-EDTA buffer and detected by staining with ethidium bromide. Strand transfer products were quantified using real-time PCR on the deproteinized DNA samples using primers 5'-CCGGCTTTCCCCGTCA AGCT and 5'-ACACCGGAGCGGATCTCG, annealing within pGEM9zf plasmid and vDNA, respectively. The real-time quantitative PCR reactions were carried out in triplicates, with 5 µM of each primer, 1 µL strand transfer product DNA (diluted 1:2500) in PowerUp SYBR-Green master mix (Thermo Fisher Scientific) and a total reaction volume of 20 µL. Relative quantities of the strand transfer products were estimated using standard curves, generated by serial dilutions of an upscaled sample with WT IN.

**TIRF microscopy and photobleaching.** All experiments were performed at room temperature in a microfluidic flow cell, functionalized with partially biotinylated polyethylene glycol (mixture of mPEG-SVA-5000 and Biotin-PEG-SVA-5000 at 50:1 mass ratio; Lysan) and assembled as previously described[101]. Prior to experiments, the biotinylated surface was coated for 10 min with 0.2 mg/mL *Streptomyces avidinii* streptavidin (Sigma-Aldrich) in PBS. All buffers and solutions were thoroughly degassed immediately before use. Flow was controlled by an automated syringe pump (Pump 11 Elite; Harvard Apparatus).

Flow cells were mounted on a Nikon Eclipse Ti inverted microscope, equipped with a 100x high numerical aperture TIRF objective (SR Apo TIRF 100×1.49 Oil; Nikon). Cy3 and Surf649 dyes were excited with 561 and 640 nm lasers (LU-N4 laser unit; Nikon), respectively, at 10% of maximum power. Emitted fluorescence was recorded using a 512 ×512 pixel, back illuminated, electron-multiplying charge-coupled-device camera (iXon DU-987, Andor Technology; 3 MHz pixel readout rate, 14-bit digitization and 300x electron multiplier gain) and a frame rate of 5 Hz. The pixel size was 160 × 160 nm.

CSC intasomes (125 µL of 4 pM solution) containing biotinylated and Cy3-labelled vDNA (prepared by annealing 5'-Cy3-CCGTGCAACACCGGAGCGGAT CTCGCA and 5'-GCTGCGAGATCCGCTCCGGTGTTGCACGG-TEG-Biotin) in Buffer A (1 M NaCl, 3 mM CaCl₂, 1 mg/mL BSA, 1 mg/mL casein, and 25 mM BisTris-HCl, pH 6.5) was drawn into the streptavidin-coated flow cell pre-equilibrated with Buffer A at a 25 µL/min flow rate and incubated for further 10 min. Excess intasome was washed off with 250 µL Buffer A at a flow rate of 50 µL/min. Next, 125 µL of a 33-nM solution of LEDGF/p75-Surf649 in Buffer B (0.5 M NaCl, 3 mM CaCl₂, 1 mg/mL BSA, 1 mg/mL casein, and 25 mM BisTris-HCl, pH 6.5) or C (0.2 M NaCl, 3 mM CaCl₂, 1 mg/mL BSA, 1 mg/mL casein, and 25 mM BisTris-HCl, pH 6.5) were introduced into the flow cell (pre-equilibrated with the same buffer) at a flow rate of 25 µL/min. After a 10-min incubation, excess LEDGF/p75-Surf649 was washed off with 250 µL of the same buffer. At this point, the following imaging sequence was implemented for three individual fields of view (512 × 512 pixels) per buffer condition: Cy3, for 5 frames, Surf649 for 3 min of continuous sampling (resulting in over 95% photobleaching within the range of the TIRF field), followed by Cy3 for 5 frames.

All data sets were initially processed with NIS Elements (Nikon) for noise reduction ("advanced denoising" at a value of 5 for both analyzed channels) and background subtraction ("rolling ball" algorithm with r = 0.48 µm). For each field of view, Cy3 (intasome) spots were detected in the 5-frame initial dataset, using "bright spot detection" (0.78 µm radius and 14.5 contrast) within NIS Elements. On average, 400 spots were detected per 512 × 512 pixel field of view. Next, for each identified intasome spot, a Surf649 (LEDGF/p75) fluorescence trace was obtained from the 3-min continuous sampling acquisition. Spots with Cy3 fluorescence intensity above 200 arbitrary units were not considered for analysis as they are likely to represent overlapping intasome complexes (the 200 arbitrary units threshold was chosen based on Cy3 intensity distribution for all analyzed data). Similarly, data with no Surf649 photobleaching steps or unstable bleaching profiles were excluded from further analysis. In order to obtain the number of Surf649 photobleaching steps per intasome, firstly, pairwise intensity differences were calculated for each photobleach trace:

$$\triangle I_{ij} = \triangle I(t_i) - \triangle I(t_j)$$

for all data pairs, for which $t_i > t_j$, followed by local maxima identification. These calculations were performed in MATLAB version R2019a (MathWorks). For each salt condition, the number of photobleaching steps per intasome was statistically analyzed for all three fields of view. Prism version 7 (GraphPad) was used for statistical analysis and data plotting.

**Cell lines and tissue culture.** Cells were cultured at 37 °C in 5% CO₂ atmosphere in Dulbecco's modified Eagle medium (DMEM, Life Technologies) supplemented with 10% heat-inactivated fetal bovine serum (FBS) and antibiotic/antimycotic solution (Sigma-Aldrich). LKO is a LEDGF-null cell line, generated via TALEN-mediated *PSIP1* gene disruption in HEK293T cells[52]. Cells additionally null for HRP2 (LHKO) were generated from LKO cells using CRISPR-Cas9 guide RNAs targeting exon 2 (5'-ACCCAACAAGTACCCCATCTTTTTC) and exon 15 (5'-CG ACCGGCAGGAGCGCGAGAGGG), resulting in deletion of most of *HDGFL2* (26 kb). Gene disruption was verified by genomic DNA sequencing, and the absence of detectable HRP2 protein was verified by immunoblotting. For ectopic expression of ovine LEDGF/p75, LHKO cells were transduced with murine leukemia virus virus-based retroviral vector pQ-OaLEDGF-IRES-PuroR and selected in the presence of 1 µg/mL puromycin. To construct this vector, a DNA fragment

encoding unaltered full-length ovine LEDGF/p75 (GenBank accession code FJ497048), PCR-amplified from a sheep peripheral blood mononuclear cell cDNA library[33] using primers 5′-GCGCATGCGGCCGCAGACACCATGACTCGCGAC TTCAAACCTGGGGACC and 5′-GGCGGGATCCCTAGTTATCTAGTGTAGA ATCCTT CAGAG, was ligated between NotI and BamHI restriction sites of pQCXIP (Clontech).

The clonal cell line CPT3 was obtained through limited dilution of CPT-Tert, ovine choroid plexus cells immortalized by co-expression of simian virus 40 large T antigen and human telomerase[53]. For genome modification of CPT3 cells, guide RNAs were prepared using CRISPR RNA targeting sheep PSIP1 or HDGFL2 genes (Supplementary Table 5) and Alt-R CRISPR-Cas9 tracrRNA (IDT) each at a final concentration of 50 μM. The ribonucleoprotein complexes were prepared by incubating 30 μM guide RNA and 24.8 μM recombinant Alt-R Cas9 Nuclease V3 (IDT) for 20 min at room temperature. CPT3 cells (1.5×10⁶) were resuspended in 70 μl SE Cell Line Nucleofector solution (Lonza) and 18–36 μl ribonucleoprotein complexes were added along with 4 μM Alt-R Cas9 electroporation enhancer (IDT) to a nucleofection chamber (Lonza). Cells were electroporated using the nucleofector program EN-138 and gently resuspended in DMEM, before plating in a six-well dish. The knockout cell lines were generated by transfection of CPT3 cells with ribonucleoprotein complexes containing crRNAs 2, 3, and 4 (LKO1 and LKO2); crRNAs 2 and 3 (LKO3); crRNAs 2 and 4 (LKO4); crRNAs 2, 3, 5, 6, 7, and 8 (LHKO1); and crRNAs 2, 4, 5, 6, 7, and 8 (LHKO2). Target sequences for CRISPR RNA are given in Supplementary Table 5. CPT-Tert cells were authenticated by sequencing of ovine cDNAs[53]. Knockout cell lines were authenticated other than by western blotting (Supplementary Figs. 8a and 9a). No other cell line authentication was performed.

**Western blotting.** The following primary antibodies were used: rabbit polyclonal anti-LEDGF/p75 (Bethyl Laboratories; product code A300-848A), rabbit polyclonal anti-HRP2 (Novus Bio; product code NBP2-47438), affinity-purified polyclonal rabbit anti-MVV capsid/p24 antibody (Thermo Fisher Scientific, custom product; see Supplementary Fig. 12 for validation data), and horseradish peroxidase-conjugated rabbit monoclonal anti-β-actin antibody clone 13E5 (Cell Signaling Technology, product code 5125). Blots probed with anti-LEDGF/p75 (diluted 1:5000 in phosphate-buffered saline, supplemented with 0.05% Tween-20 (PBST)), anti-HRP2 (diluted 1:1000 in PBST), and anti-capsid/p24 antibodies (1:5000) were developed following incubation with horseradish peroxidase-conjugated goat anti-rabbit IgG antibody (BioRad; product code 1706515; diluted 1:10,000 in PBST) or IRDye 800CW conjugated goat anti-rabbit IgG antibody (LI-COR; product code 925-32211; diluted 1:2000 in PBST) for detection by chemiluminescence, using Imager 600 RGB (GE Healthcare) following incubation with ECL prime reagent (GE Healthcare) or by fluorescence, using an Odyssey imager (LI-COR), respectively.

**MVV vectors.** The packaging construct pCAG-MV-GagPol-CTEx2[45,46] is based on MVV isolate EV1[102]. To enable mutagenesis, the construct was modified by flanking its IN-coding region with AgeI and XhoI restriction sites. To this end, a DNA fragment spanning PasI and DraIII sites of pCAG-MV-GagPol-CTEx2 was PCR-amplified and ligated between BamHI and XhoI sites of pBluescript II KS(+) (Stratagene). AgeI and XhoI restriction sites flanking the IN-coding region were introduced by silent mutagenesis, and the modified DNA fragment was ligated between PasI and DraIII sites of the packaging construct. Next, a DNA fragment encoding IN from KV1772 MVV isolate[78] was ligated between AgeI and XhoI sites of the modified packaging construct, resulting in pCAG-MV-GagPol-IN^KV1772-CTEx2. Importantly, these alterations did not affect infectivity of the MVV vector particles (Supplementary Fig. 6c) and streamlined the genetic analyses due to availability of mutant KV1772 IN expression constructs. EV1 and KV1772 INs share 86.2% amino acid sequence identity and 92% similarity. Variants of the modified packaging construct were obtained by replacing the WT IN coding region with the corresponding mutant sequences. MVV transfer vector pCVW-CG-Luc, which encodes for firefly luciferase, was constructed by overlapping PCR. Initial PCR amplicons carrying the CMV promoter and firefly luciferase were amplified from MVV transfer vector pCVW-CG and pHLLuc[103], respectively. A single linked fragment, which was produced by a second round of PCR, was digested BlpI and BglII, and then ligated with BlpI/BglII-digested pCVW-CG.

MVV vector particles were produced by transfection of HEK293T cells with four-component MVV vector system[45,46] using polyethylenimine (PEI). Briefly, HEK293T cells were plated in 15-cm tissue culture dishes to achieve ~80% confluency on the day of transfection. The cell medium was then replaced with 15 mL OptiMEM reduced serum medium (Life Technologies). PEI-DNA complexes were prepared by combining 58.5 μL 0.1% PEI (w/v in PBS, Sigma-Aldrich product number #408727) with the four-plasmid mixture containing 10.5 μg pCAG-MV-GagPol-IN^KV1772-CTEx2 (WT or mutant MVV packaging construct), 15.75 μg pCVW-CG-Luc (MVV transfer vector encoding firefly luciferase gene reporter) or pCVW-CG, which encodes for GFP, 3.5 μg pCMV-VMV-Rev (MVV Rev expression construct), and 5.25 μg pMD2.G (VSV-G expression construct, a gift from D. Trono) pre-diluted into 2.5 mL OptiMEM. Following incubation at room temperature for 15-20 min, PEI-DNA complexes were added dropwise to cells. Next day, medium was replaced with fresh DMEM supplemented with 10% FBS. Cell culture supernatant containing viral particles was

harvested 36–48 h post-transfection and pre-cleared by filtration through a 0.45-μm filter. Viral particles were pelleted at 100,000×g for 1.5 h at 25 °C in an Optima XPN ultracentrifuge using an SW 32 Ti rotor (Beckman Coulter), resuspended in 500 μL DMEM supplemented with 10% FBS, and stored in small aliquots at −80 °C. Prior to infections, viruses were treated with 0.12 U/μL TURBO DNase (Thermo Fisher Scientific) at 37 °C for 1 h. Infections of HEK293T, CPT3, and derivative cell lines were carried out in 48-well plates with the virus quantity corresponded to 0.5 mU of associated RT activity; 2 d post-infection, cells were expanded into 6-well plates, and passaged (1:2−1:10) 5 d post-infection. Cells were harvested 7 day post-infection.

**Quantification of vector particles using product-enhanced RT assay.** Virus-associated RT activity was measured using a published real-time quantitative PCR method[104] adapted for the TaqMan platform[17]. Viral particles were disrupted by combining with equal volume of lysis buffer (typically 5 μl) containing 0.8 U/μL RiboLock RNase Inhibitor (Thermo Fisher Scientific) in 50 mM KCl, 40% (v/v) glycerol, 0.25% (v/v) Triton-X100, 100 mM Tris-HCl, pH 7.4. Following incubation at room temperature for 10 min, viral lysates were diluted with 9 volumes of core buffer (20 mM KCl, 5 mM ammonium sulfate, and 20 mM Tris-HCl, pH 8.3). Each real-time PCR reaction mixture comprised 12.5 μL TaqMan Gene Expression Master Mix (Thermo Fisher Scientific), primers 5′-TCCTGCTCAACTTCCTGT CGAG and 5′- CACAGGTCAAACCTCCTAGGAATG (each at the final concentration of 0.5 μM), 0.2 μM TaqMan probe 5′-[6FAM]-CGAGACGCTAC CATGGCTATCGCTGTAG-[TAMRA], 5 U RiboLock RNase inhibitor, 100 ng phage MS2 phage RNA (Sigma-Aldrich), and 2 μL diluted viral lysate in a final volume of 25 μL. Reactions were assembled in MicroAmp Optical 96-well reaction plates and carried out in a QuantStudio-7 Flex real-time PCR instrument (Applied Biosystems). PCR conditions were as follows: 42 °C for 20 min (reverse transcription step), 50 °C for 2 min, 95 °C for 10 min (activation of the host start Taq DNA polymerase), followed by 40 cycles of PCR (denaturation at 95 °C for 15 sec and extension at 60 °C for 1 min). All PCR reactions were performed in triplicate. Standard curves were generated using recombinant HIV-1 RT (Merck Millipore). The relative viral quantities were calculated based on the standard curve generated using QuantStudio-7 systems software (Applied Biosystems).

**Luciferase assay and flow cytometry.** To measure luciferase activity, cells were washed in phosphate-buffered saline and lysed in 10 mM DTT, 10 mM EDTA, 50% glycerol, 5% triton X-100, and 125 mM Tris-HCl, pH 7.4. Following a single freeze-thaw cycle, insoluble proteins were pelleted by centrifugation at 21,000×g. Total protein content of each pre-cleared sample was determined using Pierce bicinchoninic acid assay protein assay (Thermo Fisher Scientific). To measure luciferase activity, 10 μL pre-cleared cell lysate was combined with 200 μL assay reagent containing 1 mM D-luciferin (BD Biosciences, product #556878), 3 mM adenosine triphosphate, 15 mM MgSO₄, and 30 mM sodium 4-(2-hydroxyethyl)−1-piper-azineethanesulfonate, pH 7.8. Luminescence measured using an EnVision 2102 plate reader with software version 1.13 (Perkin Elmer) was normalized by the total protein content in each precleared lysate.

Flow cytometry was used to count cells infected with GFP reporter viruses. Cells were harvested by trypsinization, fixed in 6.5% (w/v) formaldehyde in PBS and analyzed for GFP expression using a Fortessa flow cytometer (BD Biosciences). GFP was excited at 495 nm and emission was detected with a 530/30 band pass filter. Data were acquired using BD Diva software version 6.0 (BD Biosciences) and analyzed using FlowJo version 13. Live cells (population P1) were initially gated using the area plot of forward scatter (FSC-A) versus side scatter (SSC-A), to separate them from cell debris. Singe cells (population P2) were then differentiated from doublets in population P1 by gating on FSC-A vs forward scatter height (FSC-H). 10,000 single cells were analyzed for GFP expression by gating for 530/30 blue and FSC-H.

**Quantification of vDNA synthesis in infected cells.** Total DNA from infected cells was extracted and purified using DNeasy Blood and Tissue kit (Qiagen) or Quick-DNA Microprep Kit (Zymo Research) at specified times post-infection with WT or mutant MVV vectors. MVV vDNA was quantified using real-time PCR with primers 5′-GATGGTTAAGTCATAACCGCAGATG and 5′-GGTGTC TCTCTTACCTTACTT CAGG, designed to amplify a fragment within the 5′ region of the vDNA, spanning the U3-gag junction. Because the transfer vector construct pCVW-CG-Luc lacks a complete 5′ LTR (expression in transfected cells is driven by the CMV promoter), it cannot template amplification of the PCR product. Therefore, the strategy allowed the selective quantification of late reverse transcription products even in the presence of background plasmid DNA that may have persisted despite DNase treatment of vector particles prior to infection. Amplification was detected using TaqMan probe 5′-[FAM]CGTGGGGCTCGAC AAAGAATC[TAMRA]. To generate standard curves, the PCR product inserted into pCR4-TOPO vector (Thermo Fisher Scientific) was employed as template. Each real-time PCR reaction mixture comprised 10 μL TaqMan Gene Expression Master Mix (supplied as a 2x concentrate; Thermo Fisher Scientific), primers (each at the final concentration of 0.1 μM), 0.2 μM TaqMan probe, and 4 μL DNA sample (corresponding to 100 ng total DNA) in a final volume of 20 μL. Reactions were assembled in MicroAmp Optical 96-well reaction plates and carried out in a QuantStudio 7 Flex real-time PCR instrument (Applied Biosystems). PCR

conditions were as follows: 50 °C for 2 min, 95 °C for 10 min (activation of the hot start Taq DNA polymerase), followed by 40 cycles of PCR (95 °C for 15 s, 60 °C for 1 min, 72 °C for 30 s). All PCR reactions were performed in triplicate.

**Mapping MVV vector integration sites in human and ovine cells.** Genomic DNA isolated from cells 5 day post-infection with MVV vectors was processed for linker-mediated PCR, and the amplified viral LTR-chromosome junctions were sequenced as previously described[8,105]. Briefly, genomic DNA, digested with *Mse*I overnight at 37 °C, was ligated to a double-stranded DNA linker containing 5′-TA overhang overnight at 12 °C. Customized DNA linkers were used in conjunction with barcoded primers to aid in multiplexing and prevent crosstalk between samples. The first round and the nested MVV U5 primers were 5′-CTAATTCC GTGCAACACCG and 5′-AAT GAT ACG GCG ACC ACC GAG ATC TAC ACT CTT TCC CTA CAC GAC GCT CTT CCG ATC TNN NNN NCA ACA CCG GAG CGG ATC, respectively (underlined sequence represents the 6-nucleotide barcode specific to LTR primer for multiplexing). The nested PCR primers contained Illumina adaptor sequences appended at the 5′ ends. The PCR products, multiplexed on a single lane of a flow cell, were subjected to 150-bp paired-end sequencing on a HiSeq-4000 Illumina instrument (GENEWIZ, Boston, MA, USA).

Paired-end Illumina sequence reads, cropped to remove linker- and viral U5 DNA-derived sequences, were aligned with the hg38 version of human or the oviAri4 version of sheep genome using BWA version 0.7.12-r1039[106]. The results were parsed using SAMtools version 1.2[107] to extract unique, high-confidence alignments (samtools view -F 4 -F 256 -q 1). Only reads matching the host genome sequence immediately downstream of the processed vDNA U5 end were considered for further analysis. Genomic coordinates of the unique integration sites were converted to the browser extensible data format (with each interval corresponding to the middle dinucleotide of an integration site). Distributions of the integration sites with respect to various genomic features (http://genome.ucsc.edu/cgi-bin/hgTables), cLADs[55], SPADs[56] as well as local gene density and nucleotide content at the sites of integration were calculated using BEDtools version 2.25.0[108]. Expression data averaged across sheep brain tissues[109] was used to approximate gene activity in CPT-Tert cells; HEK293T gene expression data were from Gene Expression Omnibus entry GSE11892[110]. Local gene density and nucleotide content comparisons were computed using Wilcoxon rank sum test, and significance of all other comparisons were calculated with Fisher's exact or Chi-Square test in R version 4.0.4. Sequence logos were generated using WebLogo version 3[111]. HIV-1 integration sites in HEK293T and LKO cells were from published datasets[57,112]. Sequencing data and locations of mapped MVV integration sites were deposited with the Gene Expression Omnibus database under accession code GSE196042.

**Reporting summary**. Further information on research design is available in the Nature Research Reporting Summary linked to this article.

## Data availability

The data that support this study are available from the corresponding authors upon reasonable request. The final cryo-EM reconstructions were deposited with the EMDB under accession codes EMD-26322 (CSC) and EMD-14453 (STC) and fitted coordinates with the PDB under accession codes 7U32 (CSC) and 7Z1Z (STC). In addition, our Illumina sequencing data along with locations of mapped MVV integration sites were deposed with the Gene Expression Omnibus database under accession code GSE196042. Source data are provided with this paper as a Source Data file. Source data are provided with this paper.

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

## Acknowledgements

We thank Massimo Palmarini for CPT3 cells, Didier Trono for a generous gift of pMD2.G, Ron Vale for His$_{10}$-PS-SNAPf vector, Goedele Maertens for sharing the luciferase assay protocol and critical reading of the manuscript, Massimo Pizzato for advice on RT assays and nucleofection, P. Walker and A. Purkiss for computer and software support, and M. Singer for help with tissue culture. This work was funded by US National Institutes of Health grants P50 AI150481 (P.C. and A.N.E.), R01 AI070042 (A.N.E.), and U54 AI150472 (D.L.); US National Science Foundation CAREER MCB-2048095, the Margaret T. Morris Foundation, and the Hearst Foundations (D.L.); the Spanish Ministry of Science and Innovation PID2019-108850RA-I00 (JV); and the Francis Crick Institute (P.C., H.Y., and I.A.T.), which receives its core funding from Cancer Research UK (FC001061, FC001221, FC001178), the UK Medical Research Council (FC001061, FC001221, FC001178), and the Wellcome Trust (FC001061, FC001221, and FC001178).

## Author contributions

A.B.-C. optimized production of MVV STC intasomes and vitrified samples for cryo-EM; A.B.-C. and P.C. refined the STC cryo-EM structure; V.C., A.B.-C, Z,S., G.J.B., and N.C. produced recombinant proteins; V.C. conducted in vitro activity assays, studied phenotypes of the MVV vector mutants, generated CPT3-LKO and LHKO cell lines, and prepared samples for MVV integration site sequencing; V.C. and I.A.T. conducted SEC-MALLS experiments; D.T.G. and H.Y. did TIRF microscopy and photobleaching analyses; Z.S. and D.L. prepared CSC intasomes, collected cryo-EM data and refined the structure; P.K.S., A.N.E., and P.C. mapped and analyzed integration site distributions; A.N. collected STC cryo-EM data; P.C. and J.V. post-processed cryo-EM maps; V.E.P. assembled and refined the CSC and STS models; R.K.M. and D.J.G. constructed single-cycle MVV virus system; W.L. constructed luciferase reported vector; H.J.F. and E.M.P. established LKO and LHKO cell lines.

## Funding

## Competing interests

D.J.G. and R.K.M. are named as inventors on a patent application relating to the use of the MVV vectors described in this study. A.N.E. has consulted for ViiV Healthcare Co. on work unrelated to this study. No other authors declare competing interests.
