## [Peer Review File · Nature Communications]

Multivalent interactions essential for lentiviral integrase functionReviewers' Comments:

Reviewer #1:

Remarks to the Author:

This is a good paper describing an array of structural, biochemical and cell biological data on a lentiviral intasome captured in two biologically relevant states; it supersedes in quality and detail all previously available related studies.

Based on biochemical and cell biological data, the main result in the paper is that the large, hexadecameric intasome assembly is important for strand transfer activity and for integration in cells. These data were obtained by a clever selection of point mutants that allowed the assembly status of the intasome to be altered.

I am less enthusiastic about the structural and biological assays that involve the binding of LEDGF/p75 to the intasome, as I am not convinced these results can be considered novel or surprising given what is already known.

One exceptional quality of the paper is the experimental methods section that is both detailed and informative. In particular, the description of the cryo-EM work is outstanding and the described efforts to visualize the IBD domains will be of substantial value for aspiring experimental structural biologists.

I have a few minor comments.

I would like to see Fig 4S C promoted to the main figures as I believe the potential density that allowed the docking of the IBD and its interactions with the CCD dimer is important. It would be nice to see this in perhaps two orientations and the blue density should be labeled.

I would like to see a model showing how, given the results, the authors envision the intasome/nucleosome interaction. Do they believe that multiple IBDs bound to one intasome are involved or not? How would this work?

I was somewhat surprised that the paper does not describe an attempt to include a strand transfer inhibitor in the cryo-EM work, so I wonder whether there was a specific reason for this omission. I trust that relevant HIV-1 integrase strand-transfer inhibitors bind and inhibit MVV integrase as well, justifying it as a model system.

In lines 307-308 "We reasoned that the pair of positions occupied by LEDGF/p75 in our STC structure may have the highest affinity for the host factor." What would be the explanation of the differences in affinities? I thought that the IBD interacts only with the CCD dimer interface. Are there discernible differences between the available interfaces?

Later, Movie S1 is referenced in line 317, however I see no legend for this movie. For instance, what ionic strength was this taken, what are we seeing here? What was the time sequence of the experiment relative to the frames of the movie?

Reviewer #2:

Remarks to the Author:

The paper by Ballandras-Colas et al. reports cryo-EM structures of the MVV IN-DNA (intasome) complexes at improved resolution, structure-guided mutagenesis to functionally validate the observed protein-protein interaction, and single-molecule microscopy and cellular analyses to investigate how the lentivirus-specific host factor LEDGF/p75 associates with the MVV intasome (number of molecules) and impacts its integration site selection. These studies demonstrate the importance of the

hexadecameric IN assembly in lentiviral integration and significance of its interaction with LEDGF/p75, providing possible explanation for the expanded architecture of the (seemingly extraneously) large lentiviral intasomes. The experimental results presented are of high quality, carefully interpreted and clearly presented. The story is logical and easy to follow.

That said, I get an impression (perhaps naively) that the work described here represents a solid but rather incremental advancement of knowledge. The cryo-EM structures with and without the target DNA are essentially the same as those reported by the authors in 2017 in *Science*, and the mutation studies are providing functional validation of these structures. The availability of up to 16 LEDGF-IBD binding sites was illustrated in the *Science* paper as well. The roles of LEDGF/p75 and HRP-2 in lentiviral integration site selection have been studied before (e.g., PMID: 23046603).

One important question concerning lentiviral integration is whether the hexadecameric architecture is also essential for HIV/SIV. A tetrameric HIV-1 intasome structure has been reported earlier by the authors in another *Science* paper in 2017. Would it be possible to test if some of the mutations that perturb the CTD-CTD interface and the CCD-CTD linker affect HIV Integration as observed for MVV? The results in Fig. 5 show some differences in the integration site selection between MVV and HIV. Could it be explained by a possible difference in the oligomeric states of IN and resulting difference in the number of LEDGF molecules bound?

Other minor questions:

- 3D reconstruction of STC shows two copies of LEDGF-IBD whereas CSC does not have LEDGF bound. Is this significant or just reflecting differences in the sample preparation conditions?
- It is mentioned that R231 and H233 make direct DNA contacts within the expanded tDNA major groove. Do these side chains make base-contacts and play roles in determining the target sequence preference?

We thank the Reviewers for their positive evaluation and constructive criticism of our work, which helped us to improve the manuscript. The following is our point-by-point discussion of their comments.

Reviewer #1:

This is a good paper describing an array of structural, biochemical and cell biological data on a lentiviral intasome captured in two biologically relevant states; it supersedes in quality and detail all previously available related studies.

Based on biochemical and cell biological data, the main result in the paper is that the large, hexadecameric intasome assembly is important for strand transfer activity and for integration in cells. These data were obtained by a clever selection of point mutants that allowed the assembly status of the intasome to be altered.

I am less enthusiastic about the structural and biological assays that involve the binding of LEDGF/p75 to the intasome, as I am not convinced these results can be considered novel or surprising given what is already known.

Response: We respectfully disagree. Given that lentiviral intasomes contain multiple copies of integrase, there are multiple potential LEDGF/p75 binding sites. Yet, to the best of our knowledge, no one has previously reported the stoichiometry of LEDGF/p75 binding to a functional lentiviral intasome. The work in this vein is entirely novel. Our findings moreover help to clarify that the intasome may harbor a subset of comparatively high affinity LEDGF/p75 binding sites, which were resolved in our STC map and resist challenge with 1 M NaCl, and a collection of lower affinity sites that can be occupied by as many as 14 additional LEDGF/p75 molecules.

One exceptional quality of the paper is the experimental methods section that is both detailed and informative. In particular, the description of the cryo-EM work is outstanding and the described efforts to visualize the IBD domains will be of substantial value for aspiring experimental structural biologists.

Response: We thank the reviewer for recognizing the hard work that was expended to solve the new CSC and STC intasome structures, as well as the quality of the data and the resulting atomic models.

I have a few minor comments.

I would like to see Fig 4S C promoted to the main figures as I believe the potential density that allowed the docking of the IBD and its interactions with the CCD dimer is important. It would be nice to see this in perhaps two orientations and the blue density should be labeled.

Response: Thank you for these suggestions. As requested, we added a subpanel to show the cryo-EM map of the IBD region in a different orientation and labelled the blue density, although we prefer to leave the figure as supplementary (see Fig. S3c in revised

manuscript).

I would like to see a model showing how, given the results, the authors envision the intasome/nucleosome interaction. Do they believe that multiple IBDs bound to one intasome are involved or not? How would this work?

Response: Because a single nucleosome can only contain two H3 tails, the model involves interaction with multiple nucleosomes. To clarify, we have modified the relevant section of Discussion (page 18): "Simultaneous contacts with multiple H3K36Me3-containing nucleosomes can only form in locations enriched for the epigenetic mark. Such a mechanism may allow lentiviral PICs to discriminate H3K36Me3 peaks on the backdrop of noisy epigenetic landscapes..."

I was somewhat surprised that the paper does not describe an attempt to include a strand transfer inhibitor in the cryo-EM work, so I wonder whether there was a specific reason for this omission. I trust that relevant HIV-1 integrase strand-transfer inhibitors bind and inhibit MVV integrase as well, justifying it as a model system.

Response: MVV IN is indeed sensitive to strand transfer inhibitors, but two of us (Lyumkis and Cherepanov) have developed much better primate lentiviral IN systems to study drug binding (red-capped mangabey SIV IN, which is considerably closer to HIV-1 IN than is MVV IN, and yields sub-3 Å resolution cryo-EM data (PMID 32001525), as well as HIV-1 IN itself; PMID 32001521). However, unlike MVV, HIV-1 and SIV intasomes are highly heterogeneous (please refer to page 4 lines 13-21 of main text; for more details on how the lentiviral intasomes are related, please see Figs 1A and S5 in PMID 32001525). Thus, although primate lentiviral intasomes are superior to MVV for INSTI studies, their heterogeneity greatly limits applicability for structural studies that aim to assess functional IN-IN interactions, which is the goal of the present work. As discussed on page 4, MVV is the optimal primate lentiviral intasome model here.

In lines 307-308 "We reasoned that the pair of positions occupied by LEDGF/p75 in our STC structure may have the highest affinity for the host factor." What would be the explanation of the differences in affinities? I thought that the IBD interacts only with the CCD dimer interface. Are there discernible differences between the available interfaces?

Response: We thank the reviewer for these questions, which prompted us to clarify this part of the presentation. The LEDGF-IN interface involves a CCD dimer and an associated NTD (PMID 19609359, 19132083). The affinity of the interaction in the absence of the NTD is much lower (PMID 12796494). Only 2 of the 16 NTDs within the MVV intasome participate in the synaptic interface (blue and green NTDs in our figures, for example in Fig. 1a or S3c), while the remaining 14 are more loosely associated with the intasome. This may reduce the affinity of LEDGF/p75 binding at some of the sites. We have expanded our Discussion to clarify this important distinction (pages 17-18):

"The interaction of the MVV intasome with LEDGF/p75 is sensitive to buffer conditions (Fig. 4c), and only two IBDs could be located in our STC cryo-EM reconstruction (Figs S2a and S3c). Using single-molecule approaches, we showed that the intasome can bind many additional LEDGF/p75 molecules (Fig. 4c). This finding is consistent with the availability of 16 LEDGF/p75 binding sites on the IN hexadecamer (Fig. S7b), which may

display different affinities for the host factor. The IN interface with LEDGF/p75 is bipartite, involving a IN CCD dimer and associated NTD^{33,63}. Indeed, the affinity of the HIV-1 IN interaction with LEDGF/p75 in the absence of the NTD was greatly reduced⁶⁴. The two LEDGF/p75 IBDs resolved in our structure are engaged by the domain-swapped NTDs involved in the synaptic interface, and are thus stabilized within the structure via contacts with vDNA (see blue and green IN chains in Figs 1a, 1c, 1d, S3c, and S7a). The remaining 14 NTDs of the IN hexadecamer are more loosely associated with the body of the intasome, which may reduce the affinity of the corresponding sites for LEDGF/p75.”

Later, Movie S1 is referenced in line 317, however I see no legend for this movie. For instance, what ionic strength was this taken, what are we seeing here? What was the time sequence of the experiment relative to the frames of the movie?

Response: Thank you for pointing out this omission. We included the legend in the revised Supplementary Materials: "Photobleaching of the MVV intasome-LEDGF-Surf649 complexes. The recording was done in the presence of 1.0 M NaCl in the field of view presented in Fig. 4b (14.4 by 14.4 μm). The initial 5 frames show vDNA-Cy3 fluorescence (yellow), followed by the photobleaching of LEDGF-Surf649 (red) over 196 frames. The movie is presented in real time (40 s total; 5 frames per second); the scale bar is 3.0 μm ."

Reviewer #2:

The paper by Ballandras-Colas et al. reports cryo-EM structures of the MVV IN-DNA (intasome) complexes at improved resolution, structure-guided mutagenesis to functionally validate the observed protein-protein interaction, and single-molecule microscopy and cellular analyses to investigate how the lentivirus-specific host factor LEDGF/p75 associates with the MVV intasome (number of molecules) and impacts its integration site selection. These studies demonstrate the importance of the hexadecameric IN assembly in lentiviral integration and significance of its interaction with LEDGF/p75, providing possible explanation for the expanded architecture of the (seemingly extraneously) large lentiviral intasomes. The experimental results presented are of high quality, carefully interpreted and clearly presented. The story is logical and easy to follow.

That said, I get an impression (perhaps naively) that the work described here represents a solid but rather incremental advancement of knowledge. The cryo-EM structures with and without the target DNA are essentially the same as those reported by the authors in 2017 in Science, and the mutation studies are providing functional validation of these structures. The availability of up to 16 LEDGF-IBD binding sites was illustrated in the Science paper as well.

Response: In our 2017 study, we speculated that the intasome may bind as many as 16 LEDGF/p75 molecules, but we found none in the structure! Here, we refined a partially occupied complex and demonstrated that all possible sites on the intasome can at some level recruit the host factor. Based on these data and in response to Reviewer 1, we have clarified our interpretation in the revised manuscript that the intasome may very well

harbor a subset of comparatively high affinity LEDGF/p75 binding sites as well as a collection of lower affinity sites.

Our current intasome structures were refined at a much higher resolution. For example, our previous STC was refined at ~9 Å resolution, whereas the current STC is at 3.5 Å. This allowed us to discuss details of the interactions with tDNA, for example (Figs 2a and S3). The previous CSC was at ~5 Å resolution, which was insufficient to see details of the active site or IN-vDNA interactions, visible in the current 3.4-Å reconstruction (Fig. S3). Finally, the story would be incomplete without interrogation of the role the host factor plays in MVV integration. Not only have we done this in WT and genetically modified human cells, which is the norm in this field, we have extended these analyses to ovine cells, the natural host of MVV infection. Therefore, our current work greatly builds upon the previous studies and brings the work in several new directions. We honestly must agree with Reviewer 1's stance: "...it (the paper) supersedes in quality and detail all previously available related studies."

The roles of LEDGF/p75 and HRP-2 in lentiviral integration site selection have been studied before (e.g., PMID: 23046603).

Response: The paper cited by the Reviewer employed RNAi to deplete HRP-2. Beyond our work with HRP-2 knockout mice (PMID 23042676), we are unaware of any other study that has targeted HRP-2 with knockout strategies such as CRISPR. Herein we knocked out HRP-2 from both human and from ovine cells.

One important question concerning lentiviral integration is whether the hexadecameric architecture is also essential for HIV/SIV. A tetrameric HIV-1 intasome structure has been reported earlier by the authors in another Science paper in 2017.

Response: Unlike the MVV intasome, *in vitro*-assembled primate lentiviral intasomes show great degrees of heterogeneity through loss/disorder of various IN subunits and extensive stacking of the intasomes. This heterogeneity has been described in our 2020 Science paper (PMID 32001525; in particular, see Figs 1A and S5 in that publication).

HIV-1 intasomes in Passos et al 2017 were assembled using the solubility-enhancing Sso7d-IN chimera protein, rather than WT IN protein. Smaller nucleoprotein complexes were enriched through size exclusion chromatography. However, in the same study we observed much larger complexes in the presence of LEDGF/p75 and obtained a low-resolution reconstruction containing 12 HIV-1 IN molecules. There may be structural reasons for the loss of two Sso7d-IN dimers.

MVV IN is biochemically well-behaved and is thus well-suited to interrogate the effects of IN multimerization on intasome formation and function. Given both current and prior results (Passos et al 2017), we therefore argue that the hexadecamer represents a simple architectural configuration to fully complete a lentiviral intasome assembly, which should also extend to other lentiviruses.

Would it be possible to test if some of the mutations that perturb the CTD-CTD interface and the CCD-CTD linker affect HIV Integration as observed for MVV?

Response: While structurally lentiviral intasomes are clearly highly similar (as discussed above), direct comparison of HIV-1 and MVV mutants is mired by class II phenotypes, which are very common in HIV-1. Class II phenotypes are explained by a loss of IN interaction with RNA during viral maturation and disrupt the life cycle prior to reverse transcription (and therefore prior to formation of the intasome). Therefore, careful correlation of HIV-1 and MVV phenotypes will require a lot of additional experimental work and falls far outside of the scope of the current study, which focuses on validation of the expanded architecture of the MVV intasome.

The results in Fig. 5 show some differences in the integration site selection between MVV and HIV. Could it be explained by a possible difference in the oligomeric states of IN and resulting difference in the number of LEDGF molecules bound?

Response: As discussed in the paper page 18, HIV-1 accesses active, speckle-associated chromatin via CPSF6 binding, which is not shared with non-primate lentiviruses such as MVV. Thus, there are some fundamental differences between MVV and HIV-1 targeting mechanisms (CPSF6), although both viruses share LEDGF. In Fig 5, we importantly see the same trends for HIV-1 and MVV: depletion of LEDGF/p75 strongly reduced the preference for both viruses to target highly expressed genes (panels a and b) and A/T-rich regions (panels c and d). One would not expect these curves to be identical due to differences between gene expression patterns and A/T content distribution in human and ovine cells, as well as experimental noise.

Other minor questions:

- 3D reconstruction of STC shows two copies of LEDGF-IBD whereas CSC does not have LEDGF bound. Is this significant or just reflecting differences in the sample preparation conditions?

Response: Indeed, to retain LEDGF/p75 in the STC we reduced salt concentration during intasome purification: "To visualize tDNA binding and to potentially enhance host factor retainment, we assembled the STC intasome using a branched DNA construct mimicking a product of strand transfer and purified it in a buffer with reduced salt concentration" (page 6). The improved retention under low-salt conditions is fully consistent with the photobleaching data (see box/whisker plot in Fig. 4c). We expanded our Discussion to explain this point (page 17).

- It is mentioned that R231 and H233 make direct DNA contacts within the expanded tDNA major groove. Do these side chains make base-contacts and play roles in determining the target sequence preference?

Response: Arg has a flexible side chain that can make favourable major groove interactions with any single base and many combinations of bases within the major groove (PMID 11433033). The Arg231 side chain seems to be free to pack within the major groove. As such, this residue is analogous to Arg329 in PFV IN, which we showed to aid in bending tDNA (PMID 21068843). His233 does not reach far enough to make hydrogen bonds with tDNA bases. We clarified these points in Results section (pages 7-8):

“Two pairs of MVV IN Arg231 and His233 residues, located on the synaptic CTDs, along with Trp145 on the inner CCDs of the CIC, make direct contacts within the expanded

tDNA major groove (Fig. 2a). His233 is within hydrogen bonding distance of the tDNA backbone, while the side chains of Arg231 and Trp145 are in proximity with the C5 methyl group of the deoxythymidine at tDNA position -2.”